# Anyon condensation in mixed-state topological order

Ken KIKUCHI, Kah-Sen KAM and Fu-Hsiang Huang

Department of Physics, National Taiwan University, Taipei 10617, Taiwan

### Abstract

We discuss anyon condensation in mixed-state topological order. The phases were recently conjectured to be classified by pre-modular fusion categories. Just like anyon condensation in pure-state topological order, a bootstrap analysis shows condensable anyons are given by connected étale algebras. We explain how to perform generic anyon condensation including non-invertible anyons and successive condensations. Interestingly, some condensations lead to pure-state topological orders. We clarify when this happens. We also compute topological invariants of equivalence classes.

## 1 Introduction and summary

Topological order (TO) [1, 2] was found in the late 1980s. The gapped phases have peculiar properties such as robustness against local deformations, topology-dependent ground state degeneracy, and fractional statistics. All these characterizations can be concisely explained by a modern definition [3, 4, 5] as a spontaneously broken discrete higher-form symmetry [6]. Since a point links trivially with defects with codimension larger than one, local operators cannot disturb the order. On a compact space like a torus, symmetry operators can wrap nontrivial cycles giving degenerate vacua. In three-dimensional space(time), one-form symmetry operators (i.e., line operators) can have fractional spins. Due to its robustness against disturbance, TOs have been applied to, say, fault tolerant quantum computations [7].

The application in mind, one should notice one fact; the usual TOs are described by pure states because the systems are assumed to be well-isolated. However, in reality, systems interact with environments. Hence, they are described by mixed states. Therefore, if one would like to engineer realistic quantum computations with TOs, one cannot avoid studying mixed-state TOs.

Just like pure-state TOs are classified by modular fusion categories (MFCs) [8], recently, [9] conjectured that (locally-correlated) mixed-state TOs in three-dimensional space(time) are classified by pre-modular fusion categories (pre-MFCs). (The definitions are collected in section A.1.) Here, we should distinguish two mixed-state TOs: intrinsic and non-intrinsic

[10, 9]. The distinction is given as follows. By definition, a pre-MFC $\mathcal{B}$ contains transparent anyons. Their collection is denoted $\mathcal{B}'$. If there exists an MFC $\widetilde{\mathcal{B}}$ (or pure-state TO) such that $\mathcal{B}$ is given by a (Deligne tensor) product

$$\mathcal{B} \simeq \mathcal{B}' \boxtimes \widetilde{\mathcal{B}}, \tag{1.1}$$

then the mixed-state TO described by $\mathcal{B}$ is called non-intrinsic. An intuition behind this terminology is this; the mixed-state TO can be viewed as a stacking of pure-state TO $\widetilde{\mathcal{B}}$ and transparent anyons $\mathcal{B}'$. In this sense, they are boring. What is interesting is mixed-state TOs which cannot be written as products. They are called intrinsic.

In this paper, we assume the conjecture, and study the mixed-state TOs. More precisely, we take the category-theoretic (or Yoneda) perspective. Namely, in order to understand a mixed-state TO, we study its relation with other (mixed-state) TOs.

In pure-state TOs, certain relations among them are provided by the well-known procedure, anyon condensation [11, 12, 13]. The procedure achieves the following. Pick a pure-state TO described by an MFC $\mathcal{B}_1$. A 'nice' anyon $A \in \mathcal{B}_1$ can be condensed. (Here, 'nice' anyons are given by connected étale algebras in $\mathcal{B}_1$ [14].) Condensing $A$, we obtain a new pure-state TO described by another MFC $\mathcal{B}_2$. (The two phases are separated by a wall described by $\mathcal{C}$.[1]) In this way, one can obtain a new pure-state TO from an old one. Following these developments, in this paper, we study anyon condensation in mixed-state TOs.[2] Namely, we pick a mixed-state TO described by a pre-MFC $\mathcal{B}_1$ and a condensable anyon $A \in \mathcal{B}_1$. Condensing $A$, we obtain a new phase described by $\mathcal{B}_2$. We identify $\mathcal{B}_2$ and the theory $\mathcal{C}$ on the wall separating them. (In the usual pure-state TOs, particles in $\mathcal{B}_2$ and $\mathcal{C}$ are called deconfined and confined, respectively.) As we comment below, anyon condensation in mixed-state TOs allows us to compute topological invariant of such phases suggested in [9].

## 1.1 Summary of main results

Here, we summarize our results.

**Theorem.** *Condensable anyons in a mixed-state topological order $\mathcal{B}_1$ are given by connected étale algebras $A \in \mathcal{B}_1$. They give the vacuum $A = 1_{\mathcal{B}_2} \in \mathcal{B}_2$ (denoted by $\underline{0}$ in later sections) in a new phase. The new phase is given by the category $(\mathcal{B}_1)_A^0$ of dyslectic (right) A-modules. The two phases are separated by a wall given by the category $(\mathcal{B}_1)_A$ of (right) A-modules.*

---

[1]Mathematically, the new MFC $\mathcal{B}_2$ is given by the category $(\mathcal{B}_1)_A^0$ of dyslectic $A$-modules. The theory $\mathcal{C}$ on the wall is given by the category $(\mathcal{B}_1)_A$ of $A$-modules.

[2]As far as we know, anyon condensation in pre-MFC was first mentioned in the footnote 23 of [15] and studied more systematically in [16].

| Notation | Identification |
|---|---|
| Old mixed-state topological order $\mathcal{B}_1$ | Pre-modular fusion category |
| Condensable anyon $A \in \mathcal{B}_1$ | Connected étale algebra |
| New topological order $\mathcal{B}_2$ | Category $(\mathcal{B}_1)_A^0$ of dyslectic $A$-modules |
| Theory $\mathcal{C}$ on the wall | Category $(\mathcal{B}_1)_A$ of $A$-modules |

**Table 1:** Summary of results.

Interestingly, some condensations give MFCs $\mathcal{B}_2$ describing *pure-state* TOs. We clarified when this happens:

**Theorem.** *Let $\mathcal{B}_1'$ be transparent anyons in $\mathcal{B}_1$. If all transparent anyons $X \in \mathcal{B}_1'$ are bosons (i.e., they have trivial topological twists $\theta_X = 1$) and $A$ condenses all simple anyons in $\mathcal{B}_1'$, then the resulting phase is a pure-state topological order.*

This result is interesting in viewpoint of engineering; a noisy system in contact with environment can produce pure-state TO under appropriate anyon condensation. For instance, we will see a mixed-state TO described by $\mathrm{TY}(\mathbb{Z}_2 \times \mathbb{Z}_2)$ with specific conformal dimensions produces the celebrated pure-state TO, Toric Code MFC [17, 7].

The result also has implication in computation of topological invariant. A natural question in classifying mixed-state TOs is the following: when two mixed-state TOs belong to the same equivalence class? In [9], they suggested the following. Let us denote the full subcategory of transparent bosons by $\mathcal{T}_b$. If all simple anyons in $\mathcal{T}_b$ can enter a connected étale algebra, the whole $\mathcal{T}_b$ can be condensed. By the theorem, by condensing $\mathcal{T}_b$, we get either modular (when there is no transparent fermion) or super-modular fusion category (when there are transparent fermions) $\mathcal{A}^{\min}$. They called this the minimal anyon theory, and showed it is a topological invariant of an equivalence class. Developing a systematic method to condense anyons thus provides a method to compute the topological invariant. We summarized the topological invariants in our examples in the Table 2 below. As a general result, we have the

**Theorem.** *All mixed-state topological orders described by positive symmetric pre-modular fusion categories belong to the same equivalence class with $\mathcal{A}^{min} \simeq \mathrm{Vect}_{\mathbb{C}}$.*

Our examples $\mathcal{B} \simeq \mathrm{Rep}(S_3), \mathrm{Rep}(D_7), \mathrm{Rep}(S_4)$ are special cases of this general result. Relatedly, we find

**Theorem.** *If condensable anyons are transparent $A \in \mathcal{B}_1'$, then the theory on the wall is the same as the new phase $(\mathcal{B}_1)_A \simeq (\mathcal{B}_1)_A^0$. Namely, confined particles on the wall can freely move to the new phase.*

As a corollary, if condensable anyons are transparent, one can further condense anyons in $(\mathcal{B}_1)_A$ because the (whole) category admits braiding. We also study such successive anyon condensations.

We study various examples. Our examples are summarized in the following

| Pre-MFC $\mathcal{B}$ | Intrinsic? | Condensable anyon $A$ | New phase $\mathcal{B}^0_A$ | Topological invariant $\mathcal{A}^{\min}$ | Section |
|---|---|---|---|---|---|
| $\mathrm{Vec}^1_{\mathbb{Z}_Z} \boxtimes \mathrm{Fib}$ | No | $1 \oplus X$ | Fib (3.1) | Fib | 3.1 |
| $\mathrm{Rep}(D_7)$ | No | $1 \oplus X \oplus 2Y \oplus 2Z \oplus 2W$ | $\mathrm{Vect}_\mathbb{C}$ (3.2) | $\mathrm{Vect}_\mathbb{C}$ | 3.2 |
| $\mathrm{Rep}(S_4)$ | No | $1 \oplus X$ | $\mathcal{C}(\mathrm{FR}^{4,2})$ (3.3) | $\mathrm{Vect}_\mathbb{C}$ | 3.3.1 |
| | | $1 \oplus Y$ | $\mathrm{TY}(\mathbb{Z}_2 \times \mathbb{Z}_2)$ (3.4) | | 3.3.2 |
| | | $1 \oplus X \oplus 2Y$ | $\mathrm{Vec}_{\mathbb{Z}_2 \times \mathbb{Z}_2}$ or $\mathrm{Vec}_{\mathbb{Z}_4}$ (3.5,3.6) | | 3.3.3 |
| | | $1 \oplus X \oplus 2Y \oplus 3Z \oplus 3W$ | $\mathrm{Vect}_\mathbb{C}$ (3.7) | | 3.3.4 |
| $\mathrm{TY}(\mathbb{Z}_2 \times \mathbb{Z}_2)$ | Yes | $1 \oplus X$ | $\mathrm{Vec}_{\mathbb{Z}_2 \times \mathbb{Z}_2}$ or $\mathrm{Vec}_{\mathbb{Z}_4}$ (3.8) | $\begin{cases} \mathrm{Vec}^{-1}_{\mathbb{Z}_2} \boxtimes \mathrm{Vec}^{-1}_{\mathbb{Z}_2} & (h_W = \frac{1}{4}, \frac{3}{4}), \\ \mathrm{ToricCode} & (h_W = 0, \frac{1}{2}), \quad (\mathrm{mod}\ 1) \\ \mathrm{Vec}^\alpha_{\mathbb{Z}_4} & (h_W = \frac{1}{8}, \frac{3}{8}, \frac{5}{8}, \frac{7}{8}). \end{cases}$ | 3.4 |

**Table 2:** Summary of examples.

## 1.2   Procedure and examples of anyon condensation

In this subsection, we explain general procedure of anyon condensation including non-invertible anyons[3] and successive condensation. Later in the subsection, we also study simple examples to demonstrate the procedure.

The procedure consists of two steps:

1. Turn part of simple anyons in $A$ to the new vacuum $\underline{0}$,

2. Study consistencies.

In the second step, we utilize various consistencies. For example, conditions we use are

- Preservation of quantum dimensions,

- Consistency with the original fusion rules,

- Duality,

- Associativity of fusions.

We will see these in action in the later part of this subsection. Here, let us briefly see how the second step works.

---

[3]Condensation of non-invertible anyons were studied in one of the original paper [13]. For example, $0 \oplus 6 \in su(2)_{10}$ was studied in the paper.

Under anyon condensations, quantum dimensions are preserved. Since the vacuum has quantum dimension one, simple objects may or may not split into various new simple objects. For example, an invertible simple object $X$ with quantum dimension one just turns into the new vacuum

$$X \to \underline{0} \tag{1.2}$$

in order to preserve quantum dimension. If a simple object $X$ has large quantum dimension (typically no smaller than two), it splits into various new simple objects

$$X \to \underline{0} \oplus X_1 \oplus X_2 \oplus \cdots ,$$

where

$$d_X = 1 + d_{X_1} + d_{X_2} + \cdots .$$

Typically, such splittings are required to ensure the uniqueness of the vacuum in the fusion of a simple object $b_i$ and its dual $b_i^*$:

$$b_i \otimes b_i^* \cong 1 \oplus \bigoplus_{b_k \not\cong 1} N_{b_i, b_i^*}{}^k b_k. \tag{1.3}$$

The consistency may also require simple objects not in $A$, say $Y$, to split as well:

$$Y \to Y_1 \oplus Y_2 \oplus \cdots .$$

Unlike the fusion of dual pairs, simple objects in $A$ may contain more than one vacuum. For example, we will see an example $X_1 \cong \underline{0}$. In addition, consistency may force some new simple objects be identified, say $X_2 \cong Y_2$. While there are many logical possibilities as splittings or identifications, all phenomena simply follows as consequences of consistencies.

Having mentioned the quantum dimensions, let us comment on unitarity. A spherical category is called unitary if all quantum dimensions coincide with Frobenius-Perron dimensions. (See section A.1 for definitions.) Most literature on anyon condensation focused on unitary categories, however, we claim this condition is unnecessary. In fact, many anyon condensations in non-unitary MFCs were studied in [15, 16, 18, 19, 20]. Unitarity is unnecessary in our pre-MFCs either. For example, in our first example $\mathcal{B} \simeq \mathrm{Vec}^1_{\mathbb{Z}_2} \boxtimes \mathrm{Fib}$, one can condense $1 \oplus X$ in $\mathcal{B}$ with $d_{\mathrm{Fib}} = \frac{1-\sqrt{5}}{2}$. The resulting phase is still Fib, but with different quantum dimension, i.e., non-unitary Fib. Similarly, one can construct infinitely many examples of anyon condensations in non-unitary pre-MFCs as follows. Pick your favorite MFC $\mathcal{B}$ with non-unitary quantum dimension. Take a product $\mathcal{B} \boxtimes \mathrm{Vec}^1_{\mathbb{Z}_2}$. You can condense the $\mathrm{Vec}^1_{\mathbb{Z}_2}$ in the non-unitary pre-MFC.[4] With this comment, for simplicity, we limit our study to unitary pre-MFCs below.

---

[4]More generally, the $\mathbb{Z}_2$ can be generalized to a finite group $G$. It is known [21, 22] that positive symmetric $\mathrm{Rep}(G)$ can be fully condensed to get $\mathrm{Vect}_{\mathbb{C}}$.

Note that, in general, not all particles after condensation admit braiding. Particles which cannot have a well-defined braiding are called confined. (This happens if the multiple anyon types to be identified do not share the same spin factor [13].) We will see these particles are confined to the wall separating the old phase and the new phase. Unlike particles confined on the wall, particles in the new phase admit braiding.

Having explained the procedure abstractly, let us see these in concrete examples. We study two simple examples: one from the familiar pure-state TO (1) Toric Code, and another from mixed-state TO (2) $\text{Rep}(S_3)$. For the latter, we also review how to perform anyon condensation mathematically in the appendix A.3.

**Examples:**

(1) Toric code
This is a pure-state TO described by an MFC. The phase has four particles $\{1, X, Y, Z\}$ obeying fusion rules

| $\otimes$ | 1 | $X$ | $Y$ | $Z$ |
|---|---|---|---|---|
| 1 | 1 | $X$ | $Y$ | $Z$ |
| $X$ | | 1 | $Z$ | $Y$ |
| $Y$ | | | 1 | $X$ |
| $Z$ | | | | 1 |

To connect with the physics notation of Toric Code, the simple objects $Y$ and $Z$ are the vertex operator $e$ and the plaquette operator $m$, respectively. Because of the anyon permuting symmetry between $e$ and $m$, the label can be used interchangeably. The simple object $X$ is the fermion $\psi$ formed by the bound state of the vertex operator $e$ and the plaquette operator $m$. Indeed, from Table 3 and Eq.(2.28) in [18], if we restrict to the case where all quantum dimensions are positive, the available connected étale algebras are $A = 1 \oplus Y$ and $A = 1 \oplus Z$, which we recapitulate the relevant parts here:

$$A \cong \begin{cases} 1 \oplus Y & (d_X, d_Y, d_Z, h_X, h_Y, h_Z) = (1, 1, 1, \frac{1}{2}, 0, 0), \\ 1 \oplus Z & (d_X, d_Y, d_Z, h_X, h_Y, h_Z) = (1, 1, 1, \frac{1}{2}, 0, 0), \end{cases}$$

and

| Connected étale algebra $A$ | $\mathcal{B}_A$ | rank($\mathcal{B}_A$) | $\mathcal{B}_A^0$ | rank($\mathcal{B}_A^0$) |
|:---:|:---:|:---:|:---:|:---:|
| $1 \oplus Y$ | $\mathrm{Vec}_{\mathbb{Z}_2}^\alpha$ | 2 | $\mathrm{Vect}_{\mathbb{C}}$ | 1 |
| $1 \oplus Z$ | $\mathrm{Vec}_{\mathbb{Z}_2}^\alpha$ | 2 | $\mathrm{Vect}_{\mathbb{C}}$ | 1 |

,

**Table 3:** Connected étale algebra for $\mathcal{B} \simeq$ Toric Code with positive quantum dimensions

where $\mathrm{Vec}_{\mathbb{Z}_2}^\alpha$ is the usual $\mathbb{Z}_2$ category (with anomaly $\alpha$) while $\mathrm{Vect}_{\mathbb{C}}$ is the trivial category. The rank denotes the number of simple objects in a category.

Clearly, the simple objects $Y$ and $Z$ are bosons with trivial topolotical twist $\theta = e^{2\pi i h} = 1$. Besides, the conformal dimensions are also in agreement with the fermionic nature of $X$ which has conformal dimension $h_X = \frac{1}{2}$ and in turn $\theta_X = -1$. This is consistent with the fact that $e$ and $m$ operators are bosons while $\psi$ is a fermion from the usual loop picture consideration. Due to the permuting symmetry between $Y$ and $Z$, we need only consider one case of the connected étale algebra: $A = 1 \oplus Y$.

As the first step, we turn

$$1 \to \underline{0},$$
$$Y \to \underline{0},$$

where $\underline{0}$ denotes the vacuum in the condensed theory. Then the fusion rule $Y \otimes Z \cong X$ reduces to

$$\underline{0} \otimes_A Z \cong X,$$

which forces the identification

$$Z \cong \underline{X} \cong X,$$

where $\underline{X}$ is the common condensed object from $Z$ and $X$. (In later sections, we omit the underline and just write out $X$ with the understanding that they both condense to a single object.) This is an example of identification demanded by consistency. The fusion rule $X \otimes X \cong 1$ implies

$$\underline{X} \otimes_A \underline{X} \cong \underline{0}.$$

This leaves us with an MFC $\mathrm{Vec}_{\mathbb{Z}_2}^\alpha = \{\underline{0}, \underline{X}\}$ for the condensed theory.

The final step consists of the observation that $Z$ and $X$ have different spin factors: $\theta_Z = 1$ and $\theta_X = -1$. Thus, the $\underline{X}$ is confined and the final deconfined theory is just the trivial vacuum, denoted by $\mathrm{Vect}_{\mathbb{C}}$. All the results are presented in the Table 3.

(2) $\mathrm{Rep}(S_3)$

This is a mixed-state TO described by a (degenerate) pre-MFC. The phase has three particles $\{1, X, Y\}$ obeying fusion rules

| $\otimes$ | 1 | $X$ | $Y$ |
|---|---|---|---|
| 1 | 1 | $X$ | $Y$ |
| $X$ | | 1 | $Y$ |
| $Y$ | | | $1 \oplus X \oplus Y$ |

with quantum dimension

$$(d_X, d_Y) = (1, 2),$$

as well as conformal dimension

$$(h_X, h_Y) = (0, 0).$$

The connected étale algebras are given by

| Connected étale algebra $A$ | $\mathcal{B}_A \simeq \mathcal{B}_A^0$ | rank($\mathcal{B}_A$) |
|---|---|---|
| $1 \oplus X$ | $\mathrm{Vec}_{\mathbb{Z}_3}^1$ | 3 |
| $1 \oplus Y$ | $\mathrm{Vec}_{\mathbb{Z}_2}^1$ | 2 |
| $1 \oplus X \oplus 2Y$ | $\mathrm{Vect}_{\mathbb{C}}$ | 1 |

.

We have three connected étale algebras at hand and condense them consecutively. We will see an example of splitting.

1) $A = 1 \oplus X$

As a first step, we have to identify $1, X$ with the new vacuum

$$1 \to \underline{0},$$
$$X \to \underline{0}.$$

The fusion rule $Y \otimes Y \cong 1 \oplus X \oplus Y$ then becomes

$$Y \otimes_A Y \cong 2\underline{0} \oplus Y.$$

We can show $Y$ must split. Here is a proof. Assume on the contrary, it does not split, and $Y$ is simple in the new phase. Since $Y$ is self-dual, $Y^* \cong Y$, this contradicts the uniqueness of the vacuum (1.3). Therefore, $Y$ must split into two simple objects, each with quantum dimension one

$$Y \to Y_1 \oplus Y_2.$$

This is similar to the example $su(2)_4$ studied in [13]. There, they found the resulting pure-state TO is described by a $\mathbb{Z}_3$ MFC. Similarly, we find our resulting mixed-state TO is a $\mathbb{Z}_3$ degenerate pre-MFC. To see this, we study the last fusion rule $Y \otimes Y \cong 1 \oplus X \oplus Y$. After condensation, this reduces to

$$(Y_1 \oplus Y_2) \otimes_A (Y_1 \oplus Y_2) \cong 2\underline{0} \oplus Y_1 \oplus Y_2.$$

Here, since $Y^* \cong Y$, we have two logical possibilities: 1) $Y_{1,2}$ are self-dual, i.e., $Y_1^* \cong Y_1, Y_2^* \cong Y_2$, or 2) they are dual to each other, i.e., $Y_1^* \cong Y_2$. It turns out that the second possibility is correct. To show this, assume the first. The self-duality implies

$$Y_1 \otimes_A Y_1 \cong \underline{0}, \quad Y_2 \otimes_A Y_2 \cong \underline{0}.$$

Then, the fusion rule forces

$$Y_1 \otimes_A Y_2 \cong Y_1, \quad Y_2 \otimes_A Y_1 \cong Y_2,$$

or

$$Y_1 \otimes_A Y_2 \cong Y_2, \quad Y_2 \otimes_A Y_1 \cong Y_1.$$

In the first case, fuse $Y_1$ from the left to obtain a contradiction $Y_2 \cong \underline{0}$. (Here, we are using the associativity of fusion.) In the second case, fuse $Y_2$ from the right to obtain a contradiction $Y_1 \cong \underline{0}$. Therefore, $Y_{1,2}$ are dual to each other, and we learn

$$Y_1 \otimes_A Y_2 \cong \underline{0} \cong Y_2 \otimes_A Y_1.$$

Then, the fusion rule demands

$$Y_1 \otimes_A Y_1 \cong Y_1, \quad Y_2 \otimes_A Y_2 \cong Y_2,$$

or

$$Y_1 \otimes_A Y_1 \cong Y_2, \quad Y_2 \otimes_A Y_2 \cong Y_1.$$

It turns out that the second possibility is correct. To prove this, we use the same argument. Multiplying $Y_2$ from the right to the first, one finds a contradiction $Y_1 \cong \underline{0}$. Therefore, we found three simple objects $\{\underline{0}, Y_1, Y_2\}$ obeying fusion rules

| $\otimes_A$ | $\underline{0}$ | $Y_1$ | $Y_2$ |
|---|---|---|---|
| $\underline{0}$ | $\underline{0}$ | $Y_1$ | $Y_2$ |
| $Y_1$ | | $Y_2$ | $\underline{0}$ |
| $Y_2$ | | | $Y_1$ |

One recognizes the category as a $\mathbb{Z}_3$ fusion category. Since our condensable anyon is transparent, all the particles are guaranteed to have consistent braiding from the theorem in the previous subsection. Since the conformal dimensions are invariant (mod 1) under anyon condensation (A.15), we learn

$$h_{Y_1} = 0 = h_{Y_2} \pmod 1.$$

Thus, $\mathcal{B}_A \simeq \mathcal{B}_A^0$ is the degenerate $\mathbb{Z}_3$ pre-MFC.

2) $A = 1 \oplus Y$

As usual, we turn $1 \to \underline{0}$. Since $Y$ has quantum dimension two, this tells us that $Y$ must split as follows

$$Y \to \underline{0} \oplus Y_1,$$

where $d_{\mathcal{B}_A}(Y_1) = 1$. Then the fusion rule $X \otimes Y \cong Y$ reduces to

$$X \oplus (X \otimes_A Y_1) \cong \underline{0} \oplus Y_1.$$

Since we are not condensing $X$, the only option is

$$X \cong Y_1, \quad X \otimes_A Y_1 \cong \underline{0}.$$

This is an example of identification. The remaining fusion rule $Y \otimes Y \cong 1 \oplus X \oplus Y$ is automatically consistent:

$$(\underline{0} \oplus X) \otimes_A (\underline{0} \oplus X) \cong \underline{0} \oplus X \oplus \underline{0} \oplus X.$$

We find the resulting category $\mathcal{B}_A \simeq \mathcal{B}_A^0$ is the degenerate $\mathbb{Z}_2$ pre-MFC.

Since the $\underline{0} \oplus X$ is connected étale, we can further condense it in $\mathcal{B}_A \simeq \mathcal{B}_A^0$. The condensation further trivializes $X$, and we end up with the rank one MFC $\mathrm{Vect}_{\mathbb{C}}$. In the next example, we will see this successive condensation can be realized in one shot.[5]

3) $A = 1 \oplus X \oplus 2Y$

Now, as the first step, we turn

$$\begin{aligned} 1 &\to \underline{0}, \\ X &\to \underline{0}, \\ Y &\to \underline{0} \oplus Y_1. \end{aligned}$$

As the second step, we study consistency conditions imposed on these. The fusion rule $Y \otimes Y \cong 1 \oplus X \oplus Y$ leads to

$$\underline{0} \oplus 2Y_1 \oplus (Y_1 \otimes_A Y_1) \cong 3\underline{0} \oplus Y_1.$$

This tells us that the only possibility is

$$Y_1 \cong \underline{0}.$$

We are left with the trivial vacuum $\mathcal{B}_A \simeq \mathcal{B}_A^0 \simeq \mathrm{Vect}_{\mathbb{C}} = \{\underline{0}\}$.

---

[5]One could also condense $\underline{0} \oplus Y_1 \oplus Y_2$ in the previous example [16]. One can check the result is $\mathrm{Vect}_{\mathbb{C}}$. We leave the computation as an exercise for a reader.

# 2 Condensable anyons in mixed-state topological order

In this section, we study what physical requirements condensable anyons have to satisfy. We also identify which mathematical notion describes the new phase after condensation. It turns out that condensable anyons have to be connected étale. The result is the same as the usual pure-state TO [14]. In other words, non-degeneracy of pre-MFC does not play crucial role in his argument. In order to list up the physical conditions, we find it useful to see an analogy with superconductor.

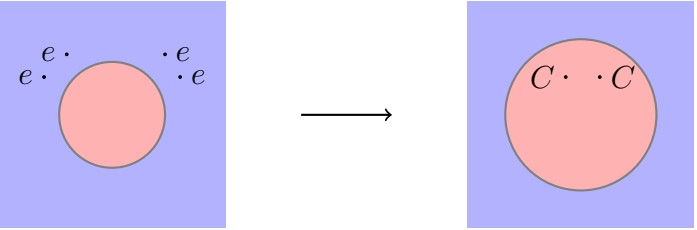

**Figure 1:** Two electrons condense to a Cooper pair in a new vacuum.

Imagine a superconducting material in a temperature higher than the critical one $T_c$. In this high temperature, electrons move around the material. We can view electrons as elementary degrees of freedom. Now, lower the temperature. When the temperature hits $T_c$, due to attractive force, electrons form Cooper pairs so as to lower energy. (See the Figure 1. The picture is a projection to a two-dimensional space.) The pairs break $U(1)$ gauge group to $\mathbb{Z}_2$. We can view the Cooper pairs as fundamental degrees of freedom at this temperature. (These are elementary fields in Ginzburg-Landau theory.) In other words, electrons are no longer fundamental degrees of freedom in this low temperature. Here, imagine we record this process, and play the video slowly. Then, we will find bubbles of true vacuum would be formulated inside the material, and they would expand to the whole superconducting material. Let us stop the video just after a bubble is created. For a sufficiently small bubble of true vacuum, it would contain only one Cooper pair. If we play the video infinitesimally, the bubble would expand slightly, and it would now contain two Cooper pairs. Here, we can perform thought experiments. Suppose we bring one Cooper pair around the other within the bubble. Since we cannot distinguish two configurations before and after the operation, the two should be equivalent. (Quantum mechanically, we 'sum up' all these possibilities.) The equivalence of two configurations explain Cooper pairs should be bosons. This line of thought experiments[6] also works in anyon condensation. As a result, we find condensable anyons have to be connected étale. To derive the consistency conditions on condensable anyons, we use the bubbles as a device. In other words, while we make use of particles in the condensed phases, one should understand analyses below in the context $\mathcal{B}_1$.

---

[6]This is just an attempt to give a physical picture to the footnote 2 of [14].

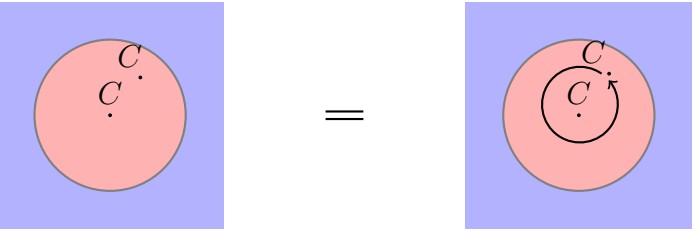

**Figure 2:** Bosonic statistics of a Cooper pair.

Now, imagine a mixed-state TO described by a pre-MFC $\mathcal{B}_1$. We denote its vacuum by $1 \in \mathcal{B}_1$. Suppose an anyon $A \in \mathcal{B}_1$ condense in the system to give another (mixed-state) TO described by $\mathcal{B}_2$.[7] Intuitively, the condensation makes $A$ 'trivial.' Therefore, just like a Cooper pair in superconductor, the condensed anyon $A$ gives the vacuum of $\mathcal{B}_2$, the 'lowest energy state.'[8] In other words, $A$ is the trivial particle in $\mathcal{B}_2$. Since the condensation selects energetically favored states, bubbles $\mathcal{B}_2$ of true vacuum $A$ would be formed in and expand through the system. Again, let us imagine we record the process, and play the video slowly. Let us stop the video just after a bubble is created. We perform various thought experiments manipulating this situation.



**Figure 3:** Unit morphism of an algebra.

As a first experiment, let us choose a point outside a bubble. Play the video so that the bubble expands and cover the point. We can view there was a trivial particle $1 \in \mathcal{B}_1$ at the point. Since the point now supports the true vacuum $A \in \mathcal{B}_2$, we should be able to turn $1 \in \mathcal{B}_1$ to $A \in \mathcal{B}_1$, which is then condensed to the vacuum of $\mathcal{B}_2$ when the bubble covers the point. Mathematically, this means an existence of unit morphism

$$u : 1 \to A \tag{2.1}$$

in $\mathcal{B}_1$.

---

[7]We will find that $\mathcal{B}_2$ can be modular describing a pure-state TO.

[8]In unitary theories, the vacuum gives the lowest energy state. However, in non-unitary theories, there may exist a state with lower energy.

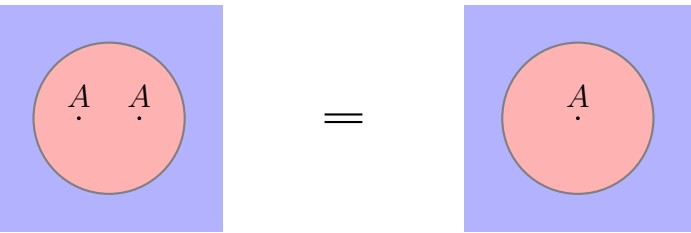

**Figure 4:** The particle $A$ is trivial.

Next, pick two arbitrary (distinct) points inside the bubble and view there are two trivial particles $A$'s at the points. We bring them together and collide them to a point (inside the bubble). Since $A \in \mathcal{B}_2$ is the trivial particle, we cannot distinguish two configurations before and after the collision. This equivalence imposes

$$A \otimes_{\mathcal{B}_2} A \cong A. \tag{2.2}$$

The collision also gives a morphism in $\mathcal{B}_1$:

$$\mu : A \otimes_{\mathcal{B}_1} A \to A. \tag{2.3}$$

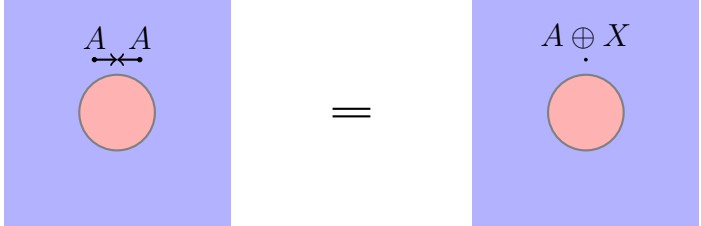

**Figure 5:** Separability of $A$.

More precise explanation is the following. After choosing two points in a bubble, rewind the tape until the bubble shrinks so small that the two points are uncovered by the bubble. Now, there are two (generically nontrivial) anyons $A \in \mathcal{B}_1$ at the points. (Just like two electrons forming a Cooper pair, $A$ is in general a composite anyon.) We bring the two anyons together and collide them. Then, play the video again until the resulting point is covered by the bubble. Since the point supported the trivial anyon $A \in \mathcal{B}_2$, the collision outside the bubble should contain $A \in \mathcal{B}_1$. The process $A \otimes_{\mathcal{B}_1} A \to A$ in $\mathcal{B}_1$ gives the multiplication morphism (2.3). We will find that the triple $(A, \mu, u)$ forms an algebra in $\mathcal{B}_1$. In general, the collision also produces other particles collectively denoted $X$. In other words, we have

$$A \otimes_{\mathcal{B}_1} A \cong A \oplus X. \tag{2.4}$$

Mathematically, this means an existence of split exact sequence

$$0 \to X \to A \otimes_{\mathcal{B}_1} A \cong A \oplus X \xrightarrow{\mu} A \to 0.$$

By the splitting lemma, this is equivalent to $\mu : A \otimes_{\mathcal{B}_1} A \to A$ admits a splitting $\tilde{\mu} : A \to A \otimes_{\mathcal{B}_1} A$ such that $\tilde{\mu} \cdot \mu = id_A$. Therefore, an algebra $(A, \mu, u)$ should be separable.

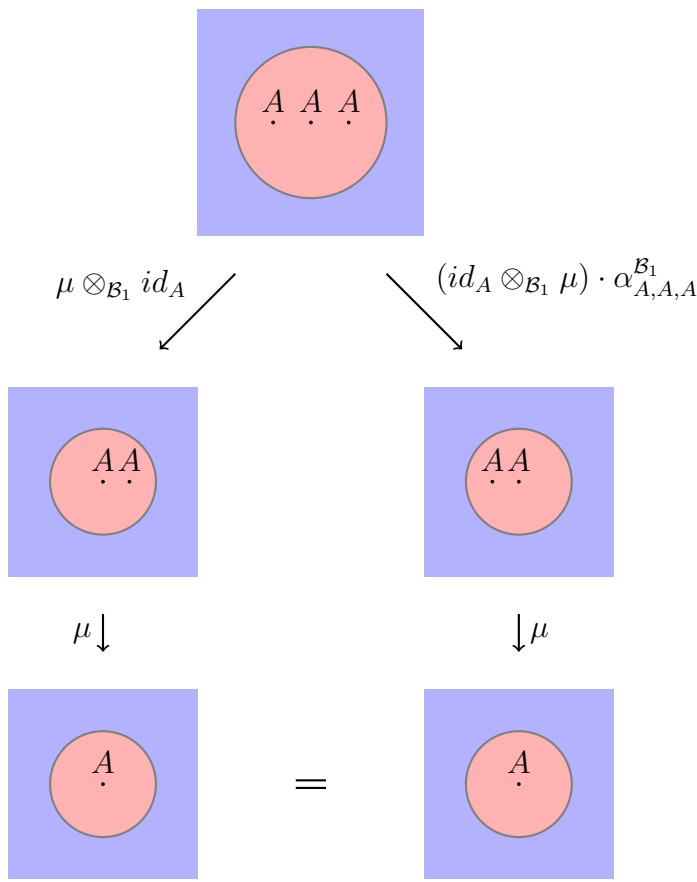

**Figure 6:** Associativity of $A$.

We can also repeat the experiment with three (distinct) points. We imagine there are three trivial particles $A \in \mathcal{B}_2$ at the points. We then bring all three $A$'s together. There are two ways to perform this operation: 1) collide the first two $A$'s first, and collide the resulting $A$ to the third $A$ later, or 2) collide the last two $A$'s first, and collide the resulting $A$ to the first $A$ later. The first process gives $\mu \cdot (\mu \otimes_{\mathcal{B}_1} id_A)$, and the second gives $\mu \cdot (id_A \otimes_{\mathcal{B}_1} \mu) \cdot \alpha^{\mathcal{B}_1}_{A,A,A}$. Again, more precisely, rewind and play the tape to get morphisms in $\mathcal{B}_1$. (Changing the order of fusions gives the additional isomorphism[9] $\alpha^{\mathcal{B}_1}_{A,A,A} : (A \otimes_{\mathcal{B}_1} A) \otimes_{\mathcal{B}_1} A \cong A \otimes_{\mathcal{B}_1} (A \otimes_{\mathcal{B}_1} A)$, the associator of $\mathcal{B}_1$.) Since we cannot distinguish the two processes, they should give the same operation:

$$\mu \cdot (\mu \otimes_{\mathcal{B}_1} id_A) = \mu \cdot (id_A \otimes_{\mathcal{B}_1} \mu) \cdot \alpha^{\mathcal{B}_1}_{A,A,A}. \tag{2.5}$$

This is nothing but the associativity axiom of $A$.

---

[9]Our notation is different from [14], but follows the standard textbook [22]. See equation (2.1).

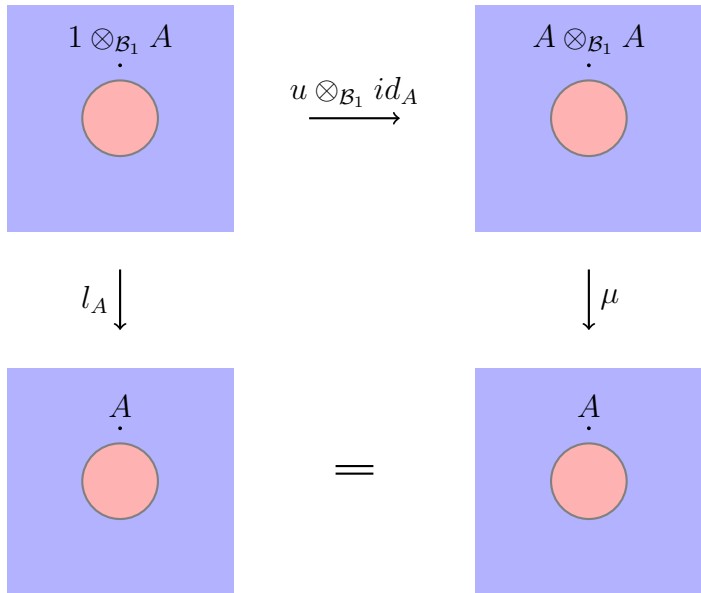

**Figure 7:** Unit axiom with the left unit constraint $l_A : 1 \otimes_{\mathcal{B}_1} A \cong A$.

To see the unit axiom of an algebra $(A, \mu, u)$, pick a point inside a bubble. We view the point supports the trivial particle $A \in \mathcal{B}_2$. Thanks to (2.2), we can also view the particle as $A \otimes_{\mathcal{B}_2} A \in \mathcal{B}_2$. Now, as usual, rewind the tape until the point is uncovered by the bubble. It should be invariant whether we view the point now supports $A \in \mathcal{B}_1$ or $A \otimes_{\mathcal{B}_1} A \in \mathcal{B}_1$. Since we can further view $A \in \mathcal{B}_1$ as fusions $1 \otimes_{\mathcal{B}_1} A \cong A \cong A \otimes_{\mathcal{B}_1} 1$, we should get various equivalent viewpoints: 1) view $1 \otimes_{\mathcal{B}_1} A$ as $A$, or 2) turn $1 \otimes_{\mathcal{B}_1} A$ to $A \otimes_{\mathcal{B}_1} A$ via (2.2), and to $A$ via (2.3). The first operation is given by $id_A \cdot l_A$, and the second by $\mu \cdot (u \otimes_{\mathcal{B}_1} id_A)$. Our inability to distinguish the two requires

$$id_A \cdot l_A = \mu \cdot (u \otimes_{\mathcal{B}_1} id_A). \tag{2.6}$$

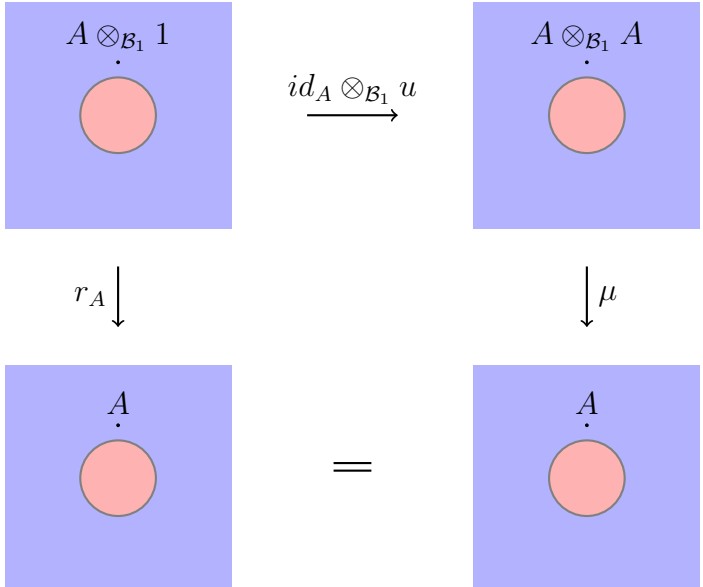

**Figure 8:** Unit axiom with the right unit constraint $r_A : A \otimes_{\mathcal{B}_1} 1 \cong A$.

Similarly, our inability to distinguish the two operations $A \otimes_{\mathcal{B}_1} 1 \xrightarrow{r_A} A$ and $A \otimes_{\mathcal{B}_1} 1 \xrightarrow{id_A \otimes_{\mathcal{B}_1} u} A \otimes_{\mathcal{B}_1} A \xrightarrow{\mu} A$ demands

$$id_A \cdot r_A = \mu \cdot (id_A \otimes_{\mathcal{B}_1} u). \tag{2.7}$$

These are nothing but the unit axioms of an algebra $(A, \mu, u)$ in $\mathcal{B}_1$. This establishes condensable anyon $A$ has to form an algebra in $\mathcal{B}_1$ together with the multiplication (2.3) and unit (2.2) morphisms.

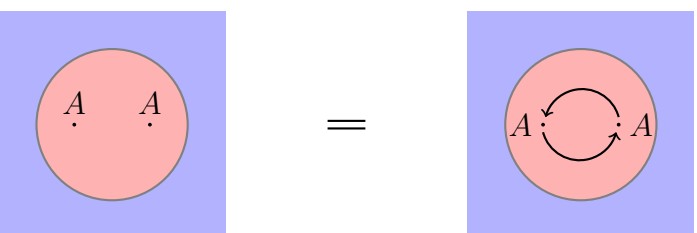

**Figure 9:** Commutativity of $A$.

As the final experiment, choose two points inside a bubble. Suppose there are two trivial particles $A \in \mathcal{B}_2$ at the points. Then, imagine to exchange two positions by dragging two trivial particles counterclockwise inside the bubble. Since $A$'s are trivial, we cannot distinguish two configurations before and after the operation. Mathematically, the exchange is provided by braiding $c_{X,Y}$. The equivalence gives

$$c_{A,A}^{\mathcal{B}_2} \cong id_{A \otimes_{\mathcal{B}_2} A}. \tag{2.8}$$

We can imagine to rewind and play the tape to obtain

$$c_{A,A}^{\mathcal{B}_1} \cong id_{A \otimes_{\mathcal{B}_1} A}.$$

This implies the algebra $A \in \mathcal{B}_1$ is commutative

$$\mu \cdot c_{A,A}^{\mathcal{B}_1} = \mu. \tag{2.9}$$

Together with the separability, this establishes $A$ should be an étale algebra. Its connectedness follows from the following observation. Since we assume the two vacua $1 \in \mathcal{B}_1$ and $A \in \mathcal{B}_2$ are unique, we expect the unit morphism (2.2) be also unique. More concrete explanation is the following. Pick one unit morphism $u : 1 \to A$ in $\mathcal{B}_1$. If your friend picks another $u' : 1 \to A$, the two unit morphisms should be related by a scalar multiplication. Mathematically, this means

$$\dim_{\mathbb{C}} \mathcal{B}_1(1, A) = 1. \tag{2.10}$$

Therefore, condensable anyons have to be connected étale algebras.

Having specified condensable anyons as connected étale algebras $A \in \mathcal{B}_1$, we next identify new phases $\mathcal{B}_2$'s after condensing them. As a result, we find $\mathcal{B}_2 \simeq (\mathcal{B}_1)_A^0$, the category of dyslectic right $A$-modules. Since the discussion is the same as [14], we will be brief.

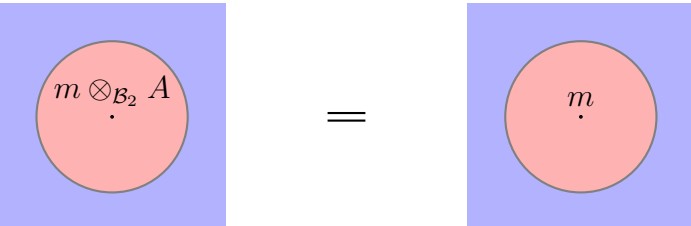

**Figure 10:** The vacuum is the unit of fusion.

Pick an arbitrary particle in the new phase, $m \in \mathcal{B}_2$. Just like a Cooper pair in superconductor, we expect $m$ is, in general, a composite particle in the old phase $\mathcal{B}_1$. It is chosen from $m \in \mathcal{B}_1$ to minimize energy in the new phase $\mathcal{B}_2$. Recall that in $\mathcal{B}_2$, $A \in \mathcal{B}_2$ is the vacuum, the trivial particle. Thus, we can view $m \in \mathcal{B}_2$ as $m \otimes_{\mathcal{B}_2} A \in \mathcal{B}_2$:

$$m \otimes_{\mathcal{B}_2} A \cong m. \tag{2.11}$$

**Figure 11:** Right action of $A$ on $m$.

By rewinding the tape, we learn there should exist a morphism in $\mathcal{B}_1$:

$$p : m \otimes_{\mathcal{B}_1} A \to m. \tag{2.12}$$

Physically, this is a selection of an energetically-favored particle $m$ from the fusion $m \otimes_{\mathcal{B}_1} A$. We will see the pair $(m, p)$ is a right $A$-module.

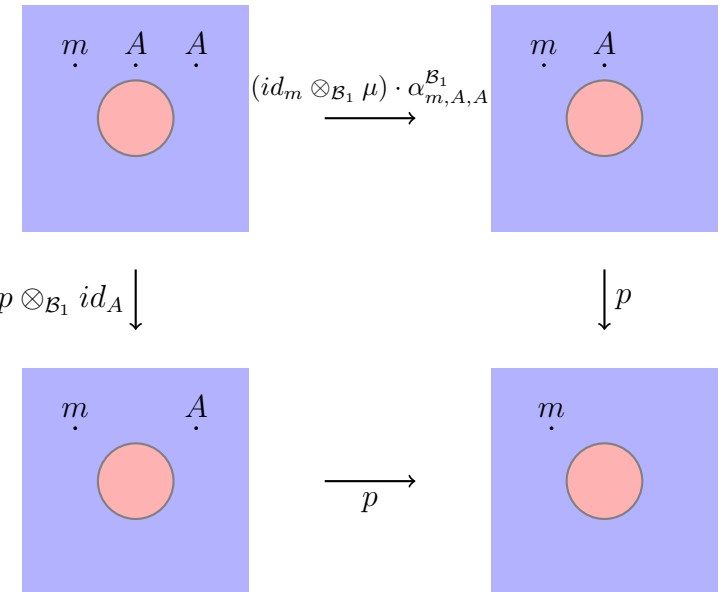

**Figure 12:** Associativity of right $A$-module.

The associativity and unit axioms of a right $A$-module $(m, p)$ can be seen as follows. Since $A \in \mathcal{B}_2$ is the trivial particle, we could equally view it as $A \otimes_{\mathcal{B}_2} A$ using (2.2). Rewinding the tape, we should get a consistent morphism $(m \otimes_{\mathcal{B}_1} A) \otimes_{\mathcal{B}_1} A \to m$ in $\mathcal{B}_1$. There are two ways: 1) first perform $p$, and perform another $p$ to the result, or 2) first change the order of fusions, fuse two $A$'s first, and perform $p$. The first procedure gives $p \cdot (p \otimes_{\mathcal{B}_1} id_A)$, and the second $p \cdot (id_m \otimes_{\mathcal{B}_1} \mu) \cdot \alpha^{\mathcal{B}_1}_{m,A,A}$. Since we cannot distinguish the two, we must have

$$p \cdot (p \otimes_{\mathcal{B}_1} id_A) = p \cdot (id_m \otimes_{\mathcal{B}_1} \mu) \cdot \alpha^{\mathcal{B}_1}_{m,A,A}. \tag{2.13}$$

This is nothing but the associativity axiom of a right $A$-module (A.8).

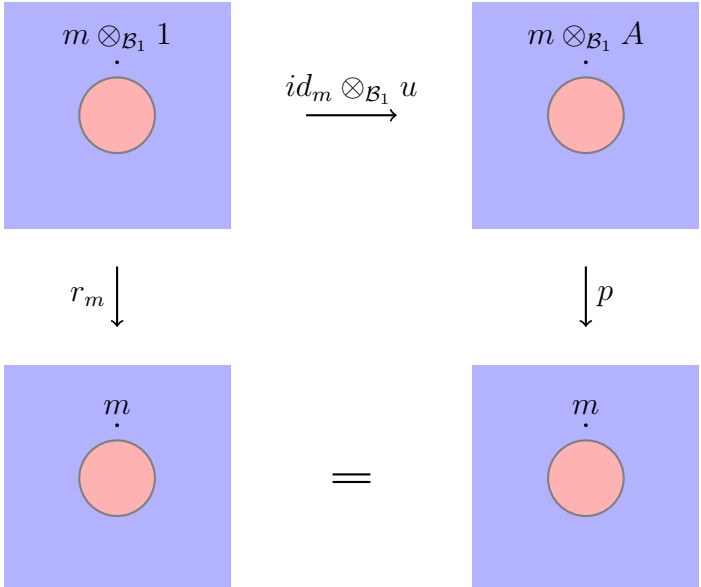

**Figure 13:** Unit axiom of right $A$-module.

The unit axiom is obtained by viewing $m \in \mathcal{B}_1$ as $m \otimes_{\mathcal{B}_1} 1$. There are two processes which turn this to $m$: 1) use (2.1) to turn it to $m \otimes_{\mathcal{B}_1} A$, and perform $p$, or 2) use the isomorphism $r_m : m \otimes_{\mathcal{B}_1} 1 \cong m$. The first method gives $p \cdot (id_m \otimes_{\mathcal{B}_1} u)$. The two cannot be distinguished, and we must have

$$p \cdot (id_m \otimes_{\mathcal{B}_1} u) = id_m \cdot r_m. \tag{2.14}$$

This is nothing but the unit axiom (A.9). Therefore, a particle $m$ – which should, rigorously speaking, be viewed as a pair $(m, p)$ – in the condensed phase $\mathcal{B}_2$ is a right $A$-module.

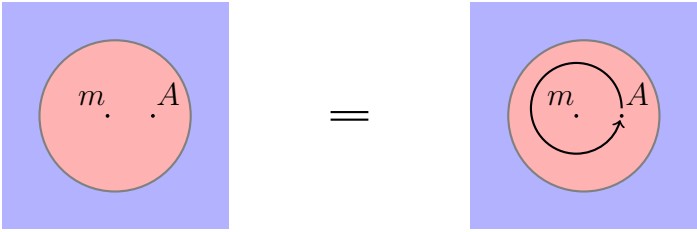

**Figure 14:** Particles in the new phase $\mathcal{B}_2$ are dyslectic right $A$-modules.

There is one additional condition on particles in $\mathcal{B}_2$. Again, pick a point in a bubble, and suppose it supports $m \in \mathcal{B}_2$. We have just seen that it can be viewed as $m \otimes_{\mathcal{B}_2} A$ in a consistent way. Now, imagine we bring the trivial particle once around $m$ counterclockwise, then perform $p$ to turn it back to $m$. Since $A$ is trivial, we cannot distinguish the two configurations before and after the operation. Again, rewind the tape, and we should get a

consistent morphism in $\mathcal{B}_1$. The operation before rotation is $p$, and that after the rotation is $p \cdot c_{A,m}^{\mathcal{B}_1} \cdot c_{m,A}^{\mathcal{B}_1}$. The two cannot be distinguished, and we get

$$p \cdot c_{A,m}^{\mathcal{B}_1} \cdot c_{m,A}^{\mathcal{B}_1} = p. \tag{2.15}$$

Namely, particles in a new phase $\mathcal{B}_2$ are dyslectic (or local) right $A$-modules obeying (A.10). Mathematically, we know they form a pre-MFC (in our setup) with fusion $\otimes_{\mathcal{B}_2}$ (or $\otimes_A$ in mathematics notation). This bootstrap analysis gives the promised identification

$$\mathcal{B}_2 \simeq (\mathcal{B}_1)_A^0. \tag{2.16}$$

The bubbles separating an old phase $\mathcal{B}_1$ and a new phase $\mathcal{B}_2 \simeq (\mathcal{B}_1)_A^0$ is identified as the category $(\mathcal{B}_1)_A$ of right $A$-modules as in [14].[10]

# 3 Examples

In this section, we study some concrete anyon condensations in pre-MFCs. Since the ambient category should be clear, we drop the subscript, and denote it by $\mathcal{B}$.

## 3.1 Example 1: $\mathcal{B} \simeq \mathbf{Vec}_{\mathbb{Z}_2}^1 \boxtimes \mathbf{Fib}$

The pre-MFC has four simple objects $\{1, X, Y, Z\}$ with fusion rules

| $\otimes$ | 1 | $X$ | $Y$ | $Z$ |
|---|---|---|---|---|
| 1 | 1 | $X$ | $Y$ | $Z$ |
| $X$ | | 1 | $Z$ | $Y$ |
| $Y$ | | | $1 \oplus Y$ | $X \oplus Z$ |
| $Z$ | | | | $1 \oplus Y$ |

.

They have

$$\mathrm{FPdim}_{\mathcal{B}}(1) = \mathrm{FPdim}_{\mathcal{B}}(X) = 1, \quad \mathrm{FPdim}_{\mathcal{B}}(Y) = \mathrm{FPdim}_{\mathcal{B}}(Z) = \zeta := \frac{1 + \sqrt{5}}{2},$$

and

$$\mathrm{FPdim}(\mathcal{B}) = 5 + \sqrt{5}.$$

---

[10]An intuitive explanation is the following. Imagine a plane. In the left half, we put the phase $\mathcal{B}_1$, and in the right half, we put the phase $\mathcal{B}_2$. In the middle, the system develops a one-dimensional wall. Particles in $\mathcal{B}_2$ can move to the wall, thus the theory describing it contains $\mathcal{B}_2$. However, two particles on the wall cannot exchange their positions without touching each other. Therefore, in general, particles on the one-dimensional wall do not admit braiding. This relaxes the condition (2.15). As a result, we identify the theory describing the bubble as $(\mathcal{B}_1)_A$.

We take a unitary pre-MFC, and they have quantum

$$d_1 = d_X = 1, \quad d_Y = d_Z = \zeta,$$

and conformal dimensions

$$(h_X, h_Y, h_Z) = (0, \pm\frac{2}{5}, \pm\frac{2}{5}) \pmod 1.$$

The pre-MFC has a nontrivial connected étale algebra

$$A \cong 1 \oplus X.$$

We condense this anyon. As the first step, we turn

$$1 \to \underline{0},$$
$$X \to \underline{0}.$$

The fusion rule $X \otimes Y \cong Z$ reduces to $\underline{0} \otimes_A Y \cong Z$. Thus, we get an identification

$$Y \cong Z.$$

Another fusion rule $Y \otimes Y \cong 1 \oplus Y$ leads to $Y \otimes_A Y \cong \underline{0} \oplus Y$. While $Y$ is self-dual, the RHS contains only one vacuum. Thus, $Y$ does not have to split. Indeed, since the quantum dimension is smaller than two, it does not split. We find $Y$ remains a simple object. The other fusion rules are automatically consistent. Therefore, the resulting phase has two simple objects $\{\underline{0}, Y\}$ obeying

| $\otimes_{A'}$ | $\underline{0}$ | $Y$ |
|:---:|:---:|:---:|
| $\underline{0}$ | $\underline{0}$ | $Y$ |
| $Y$ | | $1 \oplus Y$ |

.

We identify

$$\mathcal{B}_A \simeq \text{Fib.} \tag{3.1}$$

Since we have condensed $\mathcal{B}'$, it gives the topological invariant

$$\mathcal{A}^{\min} \simeq \text{Fib.}$$

## 3.2 Example 2: $\mathcal{B} \simeq \mathbf{Rep}(D_7)$

This pre-MFC is equipped with five simple objects $\{1, X, Y, Z, W\}$ with fusion rules

| $\otimes$ | 1 | $X$ | $Y$ | $Z$ | $W$ |
|---|---|---|---|---|---|
| 1 | 1 | $X$ | $Y$ | $Z$ | $W$ |
| $X$ | | 1 | $Y$ | $Z$ | $W$ |
| $Y$ | | | $1 \oplus X \oplus W$ | $Z \oplus W$ | $Y \oplus Z$ |
| $Z$ | | | | $1 \oplus X \oplus Y$ | $Y \oplus W$ |
| $W$ | | | | | $1 \oplus X \oplus Z$ |

Thus, simple objects have

$$\mathrm{FPdim}_{\mathcal{B}}(1) = \mathrm{FPdim}_{\mathcal{B}}(X) = 1, \quad \mathrm{FPdim}_{\mathcal{B}}(Y) = \mathrm{FPdim}_{\mathcal{B}}(Z) = \mathrm{FPdim}_{\mathcal{B}}(W) = 2,$$

and

$$\mathrm{FPdim}(\mathcal{B}) = 14.$$

We consider unitary symmetric pre-MFC. They have quantum

$$d_1 = d_X = 1, \quad d_Y = d_Z = d_W = 2$$

and conformal dimensions

$$(h_X, h_Y, h_Z, h_W) = (0, 0, 0, 0) \pmod 1.$$

The pre-MFC has a nontrivial connected étale algebra

$$A \cong 1 \oplus X \oplus 2Y \oplus 2Z \oplus 2W.$$

According to this étale algebra, all simple objects $X$, $Y$, $Z$ and $W$ are to be condensed and so we write (quantum dimension is preserved under condensation)

$$\begin{aligned} 1 &\to \underline{0}, \\ X &\to \underline{0}, \\ Y &\to \underline{0} \oplus Y_2, \\ Z &\to \underline{0} \oplus Z_2, \\ W &\to \underline{0} \oplus W_2, \end{aligned}$$

with

$$d_{\mathcal{B}_A}(Y_2) = d_{\mathcal{B}_A}(Z_2) = d_{\mathcal{B}_A}(W_2) = 1.$$

For consistency, we use fusion rule to determine what $Y_2$, $Z_2$ and $W_2$ are. The fusion rule with $X$ is automatically consistent since it acts like vacuum. The $Y \otimes Y \cong 1 \oplus X \oplus W$ reduces to

$$(\underline{0} \oplus Y_2) \otimes_A (\underline{0} \oplus Y_2) \cong \underline{0} \oplus \underline{0} \oplus \underline{0} \oplus W_2,$$

which can be simplified to

$$(Y_2 \otimes_A Y_2) \oplus 2Y_2 \cong \underline{0} \oplus \underline{0} \oplus W_2.$$

There are two possibilities $Y_2 \otimes_A Y_2 \cong \underline{0}$ or $Y_2 \otimes_A Y_2 \cong W_2$. For both cases, the result is

$$Y_2 \cong W_2 \cong \underline{0}.$$

Then $Z_2$ can be found by considering $Z \otimes Z \cong 1 \oplus X \oplus Y$, which reduces to

$$(\underline{0} \oplus Z_2) \otimes_A (\underline{0} \oplus Z_2) \cong \underline{0} \oplus \underline{0} \oplus \underline{0} \oplus \underline{0} \cong 4\underline{0},$$

This implies

$$Z_2 \cong \underline{0}.$$

To summarize, we arrive

$$1 \to \underline{0},$$
$$X \to \underline{0},$$
$$Y \to 2\underline{0},$$
$$Z \to 2\underline{0},$$
$$W \to 2\underline{0}.$$

The category of right $A$-modules is identified as

$$\mathcal{B}_A \simeq \text{Vect}_{\mathbb{C}} = \{\underline{0}\}. \tag{3.2}$$

## 3.3 Example 3: $\mathcal{B} \simeq \textbf{Rep}(S_4)$

The pre-MFC has five simple objects $\{1, X, Y, Z, W\}$ obeying fusion rules

| $\otimes$ | 1 | $X$ | $Y$ | $Z$ | $W$ |
|---|---|---|---|---|---|
| 1 | 1 | $X$ | $Y$ | $Z$ | $W$ |
| $X$ | | 1 | $Y$ | $W$ | $Z$ |
| $Y$ | | | $1 \oplus X \oplus Y$ | $Z \oplus W$ | $Z \oplus W$ |
| $Z$ | | | | $1 \oplus Y \oplus Z \oplus W$ | $X \oplus Y \oplus Z \oplus W$ |
| $W$ | | | | | $1 \oplus Y \oplus Z \oplus W$ |

.

Thus, simple objects have

$$\mathrm{FPdim}_{\mathcal{B}}(1) = 1 = \mathrm{FPdim}_{\mathcal{B}}(X), \quad \mathrm{FPdim}_{\mathcal{B}}(Y) = 2, \quad \mathrm{FPdim}_{\mathcal{B}}(Z) = 3 = \mathrm{FPdim}_{\mathcal{B}}(W),$$

and

$$\mathrm{FPdim}(\mathcal{B}) = 24.$$

We consider unitary symmetric pre-MFC. They have quantum

$$d_1 = 1 = d_X, \quad d_Y = 2, \quad d_Z = 3 = d_W$$

and conformal dimensions

$$(h_X, h_Y, h_Z, h_W) = (0, 0, 0, 0) \pmod 1.$$

The pre-MFC has nontrivial connected étale algebras[11]

$$A \cong 1 \oplus X, \ 1 \oplus Y, \ 1 \oplus X \oplus 2Y, \ 1 \oplus X \oplus 2Y \oplus 3Z \oplus 3W.$$

We condense these anyons one after another.

### 3.3.1 $A \cong 1 \oplus X$.

We have

$$1 \to \underline{0},$$
$$X \to \underline{0}.$$

The fusion $X \otimes Z \cong W$ reduces to

$$\underline{0} \otimes_A Z \cong W,$$

and we identify

$$Z \cong W.$$

We find $Y$ should split into two simple objects with quantum dimensions one. To show this, study the fusion $Y \otimes Y \cong 1 \oplus X \oplus Y$. This reduces to

$$Y \otimes_A Y \cong 2\underline{0} \oplus Y.$$

_______________________

[11]Note that

$$1 \oplus X \oplus 2Y \cong (1 \oplus X) \otimes (1 \oplus Y),$$
$$1 \oplus X \oplus 2Y \oplus 3Z \oplus 3W \cong (1 \oplus Z) \otimes (1 \oplus X \oplus 2Y) \cong (1 \oplus W) \otimes (1 \oplus X \oplus 2Y).$$

Since the RHS contains the vacuum, we learn

$$Y \cong Y^*.$$

If $Y$ were simple, the fusion contradicts the uniqueness of the vacuum (1.3). Thus, $Y$ should split into two simple objects with quantum dimensions one

$$Y \to Y_1 \oplus Y_2$$

with

$$d_{\mathcal{B}_A}(Y_1) = 1 = d_{\mathcal{B}_A}(Y_2).$$

Now, the fusion reduces to

$$(Y_1 \oplus Y_2) \otimes_A (Y_1 \oplus Y_2) \cong 2\underline{0} \oplus Y_1 \oplus Y_2.$$

The consistency analysis shows[12]

$$Y_1 \otimes_A Y_1 \cong Y_2, \quad Y_1 \otimes_A Y_2 \cong \underline{0} \cong Y_2 \otimes_A Y_1, \quad Y_2 \otimes_A Y_2 \cong Y_1.$$

Namely, $\{\underline{0}, Y_1, Y_2\}$ forms a $\mathbb{Z}_3$ (braided) fusion category. The nontrivial simple objects $Y_{1,2}$ are dual to each other, $Y_1^* \cong Y_2$. Let us see the interaction of the $\mathbb{Z}_3$ BFC and $Z \cong W$. The fusion rule $Y \otimes Z \cong Z \oplus W$ reduces to

$$(Y_1 \oplus Y_2) \otimes_A Z \cong 2Z,$$

or

$$Y_1 \otimes_A Z \cong Z \cong Y_2 \otimes_A Z.$$

Similarly for $Z \otimes Y$. Finally, $Z \otimes Z$ reduces to

$$Z \otimes_A Z \cong \underline{0} \oplus Y_1 \oplus Y_2 \oplus 2Z.$$

---

[12]Here is a proof.

We first show $Y_1 \otimes_A Y_1 \not\cong \underline{0}$. To prove this, suppose the opposite. By symmetry, we also have $Y_2 \otimes_A Y_2 \cong \underline{0}$. We are left with

$$Y_1 \otimes_A Y_2 \cong Y_1 \text{ or } Y_2.$$

For the first case, multiply $Y_1$ from the left to get a contradiction $Y_2 \cong \underline{0}$. For the second case, multiply $Y_2$ from the right to get a contradiction $Y_1 \cong \underline{0}$. Therefore, we must have

$$Y_1 \otimes_A Y_1 \not\cong \underline{0} \not\cong Y_2 \otimes_A Y_2, \quad Y_1 \otimes_A Y_2 \cong \underline{0} \cong Y_2 \otimes_A Y_1.$$

We have two possibilities

$$Y_1 \otimes_A Y_1 \cong Y_1, \quad Y_2 \otimes_A Y_2 \cong Y_2,$$

or

$$Y_1 \otimes_A Y_1 \cong Y_2, \quad Y_2 \otimes_A Y_2 \cong Y_1.$$

The first possibility is ruled out. To see this, multiply $Y_2$ from the left to the first fusion to get a contradiction $Y_1 \cong \underline{0}$. Therefore, we must have the fusion rules in the main text.

This ends the consistency analysis.

To summarize, we obtain four simple objects $\{\underline{0}, Y_1, Y_2, Z\}$ obeying fusion rules

| $\otimes_A$ | $\underline{0}$ | $Y_1$ | $Y_2$ | $Z$ |
|---|---|---|---|---|
| $\underline{0}$ | $\underline{0}$ | $Y_1$ | $Y_2$ | $Z$ |
| $Y_1$ | | $Y_2$ | $\underline{0}$ | $Z$ |
| $Y_2$ | | | $Y_1$ | $Z$ |
| $Z$ | | | | $\underline{0} \oplus Y_1 \oplus Y_2 \oplus 2Z$ |

.

Since the fusion ring has a multiplicity $N_{Z,Z}{}^{Z} = 2$, it seems the fusion category is not known. In this way, one could construct new mixed-state TOs (or even new fusion categories).

As the fusion ring has rank four and multiplicity two, let us denote it by $\mathrm{FR}^{4,2}$, and the fusion category by $\mathcal{C}(\mathrm{FR}^{4,2})$. We idenfity

$$\mathcal{B}_A \simeq \mathcal{C}(\mathrm{FR}^{4,2}). \tag{3.3}$$

### 3.3.2 $A \cong 1 \oplus Y$.

By preservation of quantum dimensions, we hve

$$1 \to \underline{0},$$
$$Y \to \underline{0} \oplus Y_2,$$

with

$$d_{\mathcal{B}_A}(Y_2) = 1.$$

The fusion $X \otimes Y \cong Y$ reduces to

$$X \otimes_A (\underline{0} \oplus Y_2) \cong \underline{0} \oplus Y_2.$$

Since the RHS contains the vacuum, $Y$ should contain the dual of $X$. As $X$ is untouched, it is different from $\underline{0}$. Hence, we must have

$$Y_2 \cong X^* \cong X.$$

$X$ swaps two simple objects $Z, W$. The fusion $Y \otimes Y \cong 1 \oplus X \oplus Y$ reduces to

$$(\underline{0} \oplus X) \otimes_A (\underline{0} \oplus X) \cong \underline{0} \oplus X \oplus \underline{0} \oplus X.$$

This is automatically consistent. The fusions $Y \otimes Z \cong Z \oplus W$ and its interchange $Z \leftrightarrow W$ are also automatically consistent. The fusion $Z \otimes Z \cong 1 \oplus Y \oplus Z \oplus W$ reduces to

$$Z \otimes_A Z \cong 2\underline{0} \oplus X \oplus Z \oplus W.$$

Since the RHS contains the vacuum, $Z$ has the conjugate of itself. If $Z$ were simple, it contradicts the uniqueness of the vacuum (1.3). Thus, $Z$ (and $W$) must split

$$Z \to Z_1 \oplus Z_2,$$
$$W \to W_1 \oplus W_2.$$

The fusion reduces to

$$(Z_1 \oplus Z_2) \otimes_A (Z_1 \oplus Z_2) \cong 2\underline{0} \oplus X \oplus Z_1 \oplus Z_2 \oplus W_1 \oplus W_2.$$

Without loss of generality, we can set

$$W_1 \cong X \otimes_A Z_1, \quad W_2 \cong X \otimes_A Z_2.$$

Thus, we have

$$d_{\mathcal{B}_A}(Z_1) = d_{\mathcal{B}_A}(W_1), \quad d_{\mathcal{B}_A}(Z_2) = d_{\mathcal{B}_A}(W_2).$$

In order to produce $2\underline{0} \oplus X$ with quantum dimension three, they must have quantum dimensions one and two:

$$d_{\mathcal{B}_A}(Z_1) = 1 = d_{\mathcal{B}_A}(W_1), \quad d_{\mathcal{B}_A}(Z_2) = 2 = d_{\mathcal{B}_A}(W_2).$$

Here, recall the fact that fusion with invertible simple object should be simple. This fact and the preservation of quantum dimensions lead to

$$Z_1 \otimes_A Z_2 \cong Z_2, \quad Z_2 \otimes_A Z_1 \cong W_2,$$

or

$$Z_1 \otimes_A Z_2 \cong W_2, \quad Z_2 \otimes_A Z_1 \cong Z_2,$$

Remembering the self-duality of $Z$, we arrive

$$Z_1 \otimes_A Z_1 \cong \underline{0}, \quad Z_2 \otimes_A Z_2 \cong \underline{0} \oplus X \oplus Z_1 \oplus W_1.$$

Another fusion $Z \otimes W \cong X \oplus Y \oplus Z \oplus W$ reduces to

$$(Z_1 \oplus Z_2) \otimes_A (W_1 \oplus W_2) \cong \underline{0} \oplus 2X \oplus Z_1 \oplus Z_2 \oplus W_1 \oplus W_2.$$

Since the RHS contains the vacuum, $W$ must contain the dual of $Z$. Comparing with the previous fusion, we conclude

$$Z_2 \cong W_2^* \cong W_2,$$

and

$$Z_1 \otimes_A W_1 \cong X, \quad W_2 \otimes_A W_1 \cong W_2.$$

Similar analysis for $W \otimes W$ leads to

$$W_1 \otimes_A W_1 \cong \underline{0}.$$

To sum up, we obtain five simple objects $\{\underline{0}, X, Z_1, W_1, W_2\}$ obeying fusion rules

| $\otimes_A$ | $\underline{0}$ | $X$ | $Z_1$ | $W_1$ | $W_2$ |
|---|---|---|---|---|---|
| $\underline{0}$ | $\underline{0}$ | $X$ | $Z_1$ | $W_1$ | $W_2$ |
| $X$ | | $\underline{0}$ | $W_1$ | $Z_1$ | $W_2$ |
| $Z_1$ | | | $\underline{0}$ | $X$ | $W_2$ |
| $W_1$ | | | | $\underline{0}$ | $W_2$ |
| $W_1$ | | | | | $\underline{0} \oplus X \oplus Z_1 \oplus W_1$ |

One recognizes

$$\mathcal{B}_A \simeq \mathrm{TY}(\mathbb{Z}_2 \times \mathbb{Z}_2), \tag{3.4}$$

a $\mathbb{Z}_2 \times \mathbb{Z}_2$ Tambara-Yamagami category.

### 3.3.3 $A \cong 1 \oplus X \oplus 2Y$.

Since quantum dimensions are preserved, we have

$$1 \to \underline{0},$$
$$X \to \underline{0},$$
$$Y \to \underline{0} \oplus Y_2,$$

with

$$d_{\mathcal{B}_A}(Y_2) = 1.$$

The fusion $X \otimes Z \cong W$ reduces to

$$\underline{0} \otimes_A Z \cong W.$$

We thus identify

$$W \cong Z.$$

The fusion $Y \otimes Y \cong 1 \oplus X \oplus Y$ reduces to

$$(\underline{0} \oplus Y_2) \otimes_A (\underline{0} \oplus Y_2) \cong 3\underline{0} \oplus Y_2.$$

By preservation of quantum dimensions, we must have

$$Y_2 \cong \underline{0}.$$

The fusions $Y \otimes Z \cong Z \oplus W$ and $Y \otimes W \cong Z \oplus W$ are automatically satisfied. The fusion $Z \otimes Z \cong 1 \oplus Y \oplus Z \oplus W$ reduces to

$$Z \otimes_A Z \cong 3\underline{0} \oplus 2Z.$$

Since the RHS contains the vacuum, $Z$ must contain its dual. Then, $Z$ cannot be simple:

$$Z \to Z_1 \oplus Z_2.$$

(It may further split. Indeed, we will see it does.) For its self-fusion to produce three vacua, we must have

$$d_{\mathcal{B}_A}(Z_1) = 1, \quad d_{\mathcal{B}_A}(Z_2) = 2.$$

Simplicity of $Z_1 \otimes_A Z_2, Z_2 \otimes_A Z_1$ gives

$$Z_1 \otimes_A Z_2 \cong Z_2 \cong Z_2 \otimes_A Z_1.$$

The fusion above reduces to

$$Z_1 \otimes_A Z_1 \oplus Z_2 \otimes_A Z_2 \cong 3\underline{0} \oplus 2Z_1.$$

Regardless of $Z_1 \otimes_A Z_1$, this implies $Z_2 \otimes_A Z_2$ contains more than one vacua. Thus, $Z_2$ must further split into two simple objects with quantum dimensions one:

$$Z_2 \to Z_2' \oplus Z_2'',$$

with

$$d_{\mathcal{B}_A}(Z_2') = 1 = d_{\mathcal{B}_A}(Z_2'').$$

The fusion now reduced to

$$Z_1 \otimes_A Z_1 \oplus (Z_2' \oplus Z_2'') \otimes_A (Z_2' \oplus Z_2'') \cong 3\underline{0} \oplus 2Z_1.$$

Since $Z_1, Z_2', Z_2''$ enters symmetrically and one of their self-fusions should be the vacuum, we set

$$Z_1 \otimes_A Z_1 \cong \underline{0}.$$

We are left with

$$(Z_2' \oplus Z_2'') \otimes_A (Z_2' \oplus Z_2'') \cong 2\underline{0} \oplus 2Z_1.$$

Here, we have two possibilities: 1) $Z_2' \otimes_A Z_2' \cong \underline{0} \cong Z_2'' \otimes_A Z_2''$, or 2) $Z_2' \otimes_A Z_2' \cong Z_1 \cong Z_2'' \otimes_A Z_2''$. For the first case, we must have

$$Z_2' \otimes_A Z_2'' \cong Z_1 \cong Z_2'' \otimes_A Z_2'.$$

The consistency reduces $Z_1 \otimes_A (Z_2' \oplus Z_2'') \cong Z_2' \oplus Z_2''$ to[13]

$$Z_1 \otimes_A Z_2' \cong Z_2'', \quad Z_1 \otimes_A Z_2'' \cong Z_2'.$$

For the second case, we have

$$Z_2' \otimes_A Z_2'' \cong \underline{0} \cong Z_2'' \otimes_A Z_2'.$$

The consistency similarly reduces $Z_1 \otimes_A (Z_2' \oplus Z_2'') \cong Z_2' \oplus Z_2''$ to[14]

$$Z_1 \otimes_A Z_2' \cong Z_2'', \quad Z_1 \otimes_A Z_2'' \cong Z_2'.$$

This ends our analysis.

To summarize, we obtain four simple objects $\{\underline{0}, Z_1, Z_2', Z_2''\}$. Their fusions depend on, say, $Z_2' \otimes_A Z_2'$. When it is trivial, we have

| $\otimes_A$ | $\underline{0}$ | $Z_1$ | $Z_2'$ | $Z_2''$ |
|---|---|---|---|---|
| $\underline{0}$ | $\underline{0}$ | $Z_1$ | $Z_2'$ | $Z_2''$ |
| $Z_1$ | | $\underline{0}$ | $Z_2''$ | $Z_2'$ |
| $Z_2'$ | | | $\underline{0}$ | $Z_1$ |
| $Z_2''$ | | | | $\underline{0}$ |

and when it is $Z_1$, we have

| $\otimes_A$ | $\underline{0}$ | $Z_1$ | $Z_2'$ | $Z_2''$ |
|---|---|---|---|---|
| $\underline{0}$ | $\underline{0}$ | $Z_1$ | $Z_2'$ | $Z_2''$ |
| $Z_1$ | | $\underline{0}$ | $Z_2''$ | $Z_2'$ |
| $Z_2'$ | | | $Z_1$ | $\underline{0}$ |
| $Z_2''$ | | | | $Z_1$ |

They are identified as either

$$\mathcal{B}_A \simeq \mathrm{Vec}_{\mathbb{Z}_2 \times \mathbb{Z}_2}, \tag{3.5}$$

or

$$\mathcal{B}_A \simeq \mathrm{Vec}_{\mathbb{Z}_4}, \tag{3.6}$$

respectively.

---

[13]To show this, assume the opposite $Z_1 \otimes_A Z_2' \cong Z_2'$, and multiply $Z_2'$ from the right to get a contradiction $Z_1 \cong \underline{0}$.

[14]To prove this, assume the opposite $Z_1 \otimes_A Z_2' \cong Z_2'$, and multiply $Z_2''$ from the right to get a contradiction $Z_1 \cong \underline{0}$.

**3.3.4** $A \cong 1 \oplus X \oplus 2Y \oplus 2Z \oplus 2W$**.**

By preservation of quantum dimensions, we have

$$1 \to \underline{0},$$
$$X \to \underline{0},$$
$$Y \to \underline{0} \oplus Y_2,$$
$$Z \to \underline{0} \oplus Z_2,$$
$$W \to \underline{0} \oplus W_2,$$

with

$$d_{\mathcal{B}_A}(Y_2) = 1, \quad d_{\mathcal{B}_A}(Z_2) = 2 = d_{\mathcal{B}_A}(W_2).$$

From the quantum dimensions, we learn $Z_2, W_2$ may further split. To determine their fates, we study consistency with the original fusion rules. The $X \otimes Y \cong Y$ is automatically consistent. The $X \otimes Z \cong W$ reduces to

$$\underline{0} \otimes_A (\underline{0} \oplus Z_2) \cong \underline{0} \oplus W_2,$$

and demands

$$Z_2 \cong W_2.$$

The $Y \otimes Y \cong 1 \oplus X \oplus Y$ reduces to

$$(\underline{0} \oplus Y_2) \otimes_A (\underline{0} \oplus Y_2) \cong \underline{0} \oplus \underline{0} \oplus \underline{0} \oplus Y_2,$$

and forces

$$Y_2 \cong \underline{0}.$$

The $Y \otimes Z \cong Z \oplus W$ and its interchange $Z \leftrightarrow W$ are automatically consistent. Finally, the $Z \otimes Z \cong 1 \oplus Y \oplus Z \oplus W$ reduces to

$$(\underline{0} \oplus Z_2) \otimes_A (\underline{0} \oplus Z_2) \cong \underline{0} \oplus 2\underline{0} \oplus 2(\underline{0} \oplus Z_2).$$

This forces

$$Z_2 \otimes_A Z_2 \cong 4\underline{0}.$$

Since the RHS contains the vacuum, this fusion rule implies $Z_2$ is its self-dual, $Z_2^* \cong Z_2$. Then, we can argue $Z_2$ should split into two simple objects with quantum dimensions one. To prove this, assume the contrary; suppose $Z_2$ be simple. Then, the fusion rule contradicts the uniqueness of the vacuum (1.3). Thus, we must have

$$Z_2 \cong Z_2' \oplus Z_2''$$

with

$$d_{\mathcal{B}_A}(Z_2') = 1 = d_{\mathcal{B}_A}(Z_2'').$$

The fusion rule above reduces to

$$(Z_2' \oplus Z_2'') \otimes_A (Z_2' \oplus Z_2'') \cong 4\underline{0},$$

or

$$Z_2' \otimes_A Z_2' \cong Z_2' \otimes_A Z_2'' \cong Z_2'' \otimes_A Z_2' \cong Z_2'' \otimes_A Z_2'' \cong \underline{0}.$$

Multiplying $Z_2'$ from the left to $Z_2' \otimes_A Z_2'' \cong \underline{0}$, we get

$$Z_2' \cong Z_2''.$$

If $Z_2'$ were not isomorphic to the vacuum $\underline{0}$, the category $\mathcal{B}_A$ of right $A$-modules would be the $\mathbb{Z}_2$ fusion category $\{\underline{0}, Z_2'\}$. It has

$$\mathrm{FPdim}(\mathrm{Vec}_{\mathbb{Z}_2}^\alpha) = 2,$$

and contradicts

$$\mathrm{FPdim}(\mathcal{B}_A) = \frac{\mathrm{FPdim}(\mathcal{B})}{\mathrm{FPdim}_{\mathcal{B}}(A)} = 1.$$

Therefore, we must have

$$Z_2' \cong \underline{0}.$$

This is the final consistency condition. To summarize, we arrive

$$1 \to \underline{0},$$
$$X \to \underline{0},$$
$$Y \to 2\underline{0},$$
$$Z \to 3\underline{0},$$
$$W \to 3\underline{0}.$$

The category of right $A$-modules is identified as

$$\mathcal{B}_A \simeq \mathrm{Vect}_{\mathbb{C}} = \{\underline{0}\}. \tag{3.7}$$

### 3.3.5 Successive condensation.

In this section, we study two successive condensations. In the previous sections, we have condensed $1 \oplus X, 1 \oplus Y \in \mathrm{Rep}(S_4)$. We condense the other one from the result.

Condensing $1 \oplus X \in \mathrm{Rep}(S_4)$ in the section 3.3.1, we found $\mathcal{B}_A \simeq \mathcal{C}(\mathrm{FR}^{4,2})$. We condense $1 \oplus Y$, which splits into

$$A' \cong 1 \oplus Y_1 \oplus Y_2 \in \mathcal{B}_A,$$

where $Y_1$ and $Y_2$ have quantum dimensions one. In the first step, we turn

$$1 \to \underline{0},$$
$$Y_1 \to \underline{0},$$
$$Y_2 \to \underline{0}.$$

The $\mathcal{B}_A$ has fusion $Z \otimes_A Z \cong \underline{0} \oplus Y_1 \oplus Y_2 \oplus 2Z$, where $Z$ has quantum dimension three. After condensation, we obtain

$$Z \otimes_{A'} Z \cong \underline{0} \oplus \underline{0} \oplus \underline{0} \oplus 2Z,$$

The self-dual object $Z$ is not simple, otherwise, the RHS should only has one vacuum. Thus, $Z$ splits into $Z_1$ and $Z_2$ with quantum dimensions one and two. Plugging it back to the fusion product, we get

$$Z_1 \otimes_{A'} Z_1 \oplus Z_1 \otimes_{A'} Z_2 \oplus Z_2 \otimes_{A'} Z_1 \oplus Z_2 \otimes_{A'} Z_2 \cong \underline{0} \oplus \underline{0} \oplus \underline{0} \oplus 2Z_1 \oplus 2Z_2.$$

We know that $Z_1$ is an invertible object, therefore, for an arbitrarily simple object $b$, $Z_1 \otimes_{A'} b$ and $b \otimes_{A'} Z_1$ should be simple. Hence, $Z_1 \otimes_{A'} Z_2$ and $Z_2 \otimes_{A'} Z_1$ are simple objects with quantum dimensions two. They can only be $Z_2$. Then, we obtain

$$Z_1 \otimes_{A'} Z_1 \cong \underline{0},$$
$$Z_1 \otimes_{A'} Z_2 \cong Z_2 \cong Z_2 \otimes_{A'} Z_1,$$
$$Z_2 \otimes_{A'} Z_2 \cong 2\underline{0} \oplus 2Z_1.$$

The self-duality of $Z$ implies $Z_1$ and $Z_2$ are both self-dual. This implies $Z_2$ further splits. To see this, suppose $Z_2$ were simple. Then, from the self-duality, its fusion with itself should contain only one vacuum, a contradiction. Hence, $Z_2$ is not simple. Therefore, $Z_2$ splits into $Z_2'$ and $Z_2''$ with quantum dimensions one:

$$Z_2 \to Z_2' \oplus Z_2''.$$

Substituting this back to the fusion product again, we get

$$Z_2' \otimes_{A'} Z_2' \oplus Z_2' \otimes_{A'} Z_2'' \oplus Z_2'' \otimes_{A'} Z_2' \oplus Z_2'' \otimes_{A'} Z_2'' \cong 2\underline{0} \oplus 2Z_1.$$

Since $Z_2$ is self-dual $Z_2^* \cong Z_2$, $Z_2'$ and $Z_2''$ are self-dual or dual to each other. For the first case, it leads to

$$Z_2' \otimes_{A'} Z_2' \cong \underline{0} \cong Z_2'' \otimes_{A'} Z_2''.$$

Then, the fusion rule demands

$$Z_2' \otimes_{A'} Z_2'' \cong Z_1 \cong Z_2'' \otimes_{A'} Z_2'.$$

For the other case, we have $Z_2'^* \cong Z_2''$, and

$$Z_2' \otimes_{A'} Z_2'' \cong \underline{0} \cong Z_2'' \otimes_{A'} Z_2', \quad Z_2' \otimes_{A'} Z_2' \cong Z_1 \cong Z_2'' \otimes_{A'} Z_2''.$$

We found the four simple objects $\{\underline{0}, Z_1, Z_2', Z_2''\}$ obeying

| $\otimes_A'$ | $\underline{0}$ | $Z_1$ | $Z_2'$ | $Z_2''$ |
|---|---|---|---|---|
| $\underline{0}$ | $\underline{0}$ | $Z_1$ | $Z_2'$ | $Z_2''$ |
| $Z_1$ | | $\underline{0}$ | $Z_2''$ | $Z_2'$ |
| $Z_2'$ | | | $\underline{0}$ | $Z_1$ |
| $Z_2''$ | | | | $\underline{0}$ |

,

or

| $\otimes_A'$ | $\underline{0}$ | $Z_1$ | $Z_2'$ | $Z_2''$ |
|---|---|---|---|---|
| $\underline{0}$ | $\underline{0}$ | $Z_1$ | $Z_2'$ | $Z_2''$ |
| $Z_1$ | | $\underline{0}$ | $Z_2''$ | $Z_2'$ |
| $Z_2'$ | | | $Z_1$ | $\underline{0}$ |
| $Z_2''$ | | | | $Z_1$ |

.

Hence, we get the same result as in section 3.3.3. They are identified as either

$$\mathcal{B}_A \simeq \mathrm{Vec}_{\mathbb{Z}_2 \times \mathbb{Z}_2},$$

or

$$\mathcal{B}_A \simeq \mathrm{Vec}_{\mathbb{Z}_4},$$

respectively.

Next, we consider another path,

$$\mathcal{B} \xrightarrow{1 \oplus Y} \mathcal{B}_A' \xrightarrow{1 \oplus X} \text{result}.$$

From the result in section 3.3.2, take

$$A' \cong 1 \oplus X \in \mathcal{B}_A,$$

where $X$ has quantum one. We turn

$$1 \to \underline{0},$$
$$X \to \underline{0}.$$

From the fusion rule, we have

$$X \otimes_A Z_1 \cong W_1, \quad X \otimes_A W_1 \cong Z_1, \quad Z_1 \otimes_A W_1 \cong X.$$

After condensation, these reduce to

$$\underline{0} \otimes_{A'} Z_1 \cong W_1, \quad \underline{0} \otimes_{A'} W_1 \cong Z_1, \quad Z_1 \otimes_{A'} W_1 \cong \underline{0}.$$

It leads to

$$Z_1 \cong W_1, \quad Z_1 \otimes_{A'} Z_1 \cong 0.$$

Another fusion rule $W_2 \otimes_A W_2 \cong \underline{0} \oplus X \oplus Z_1 \oplus W_1$ reduces to

$$W_2 \otimes_{A'} W_2 \cong 2\underline{0} \oplus 2Z_1.$$

Just as before, since $W_2$ is self-dual, it is not simple. Hence, $W_2$ splits into $W_2'$ and $W_2''$ with quantum dimensions one. Putting this back into the fusion product, we obtain

$$W_2' \otimes_{A'} W_2' \oplus W_2' \otimes_{A'} W_2'' \oplus W_2'' \otimes_{A'} W_2' \oplus W_2'' \otimes_{A'} W_2'' \cong 2\underline{0} \oplus 2Z_1.$$

Exactly the same as in the previous condensation, there are two possibilities: 1) $W_2'$ and $W_2''$ are both self-dual, or 2) they are dual to each other. In the first case, we have

$$W_2' \otimes_{A'} W_2' \cong \underline{0} \cong W_2'' \otimes_{A'} W_2'',$$

and

$$W_2' \otimes_{A'} W_2'' \cong Z_1 \cong W_2'' \otimes_{A'} W_2'.$$

In the second case, we have

$$W_2' \otimes_{A'} W_2'' \cong \underline{0} \simeq W_2'' \otimes_{A'} W_2',$$

and

$$W_2' \otimes_{A'} W_2' \cong Z_1 \cong W_2'' \otimes_{A'} W_2''.$$

The remaining fusion $Z_1 \otimes_A W_2 \cong W_2$ reduces to

$$Z_1 \otimes_{A'} (W_2' \oplus W_2'') \cong W_2' \oplus W_2''.$$

In both cases, consistency with associativity[15] leads to

$$Z_1 \otimes_{A'} W_2' \cong W_2'', \quad Z_1 \otimes_{A'} W_2'' \cong W_2'.$$

---

[15]There are two logical possibilities:

$$Z_1 \otimes_{A'} W_2' \cong W_2', \quad Z_1 \otimes_{A'} W_2'' \cong W_2'',$$

or

$$Z_1 \otimes_{A'} W_2' \cong W_2'', \quad Z_1 \otimes_{A'} W_2'' \cong W_2'.$$

The first possibility is inconsistent with associativity. To show this, when $W_2', W_2''$ are self-dual, multiply $W_2'$ from the right to get $Z_1 \cong \underline{0}$, a contradiction. When they are dual to each other, multiply $W_2''$ from the right to get the same contradiction.

To summarize, we have four simple objects $\{\underline{0}, Z_1, W_2', W_2''\}$ obeying

| $\otimes_{A'}$ | $\underline{0}$ | $Z_1$ | $W_2'$ | $W_2''$ |
|---|---|---|---|---|
| $\underline{0}$ | $\underline{0}$ | $Z_1$ | $W_2'$ | $W_2''$ |
| $Z_1$ | | $\underline{0}$ | $W_2''$ | $W_2'$ |
| $W_2'$ | | | $\underline{0}$ | $Z_1$ |
| $W_2''$ | | | | $\underline{0}$ |

,

or

| $\otimes_{A'}$ | $\underline{0}$ | $Z_1$ | $W_2'$ | $W_2''$ |
|---|---|---|---|---|
| $\underline{0}$ | $\underline{0}$ | $Z_1$ | $W_2'$ | $W_2''$ |
| $Z_1$ | | $\underline{0}$ | $W_2''$ | $W_2'$ |
| $W_2'$ | | | $Z_1$ | $\underline{0}$ |
| $W_2''$ | | | | $Z_1$ |

.

Therefore, we get the same result as in the previous successive condensation and section 3.3.3. They are identified as either

$$\mathcal{B}_A \simeq \mathrm{Vec}_{\mathbb{Z}_2 \times \mathbb{Z}_2},$$

or

$$\mathcal{B}_A \simeq \mathrm{Vec}_{\mathbb{Z}_4},$$

respectively. We found no matter which paths we choose, we will get the same result.

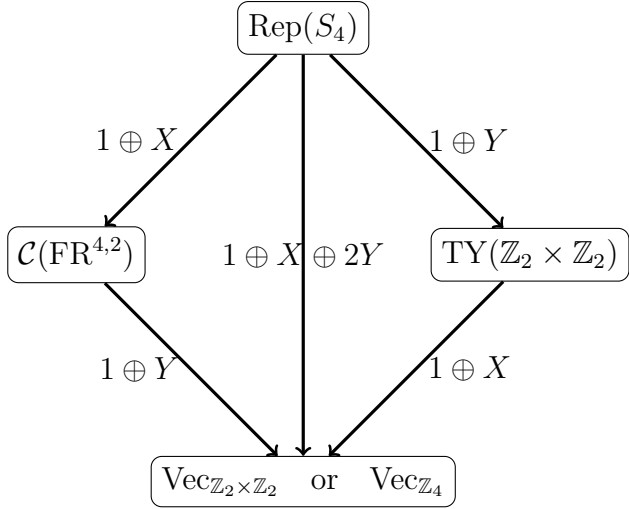

## 3.4 Example 4: $\mathcal{B} \simeq \mathbf{TY}(\mathbb{Z}_2 \times \mathbb{Z}_2)$

The pre-MFC has five simple objects $\{1, X, Y, Z, W\}$ obeying fusion rules

| $\otimes$ | 1 | $X$ | $Y$ | $Z$ | $W$ |
|-----------|---|-----|-----|-----|-----|
| 1 | 1 | $X$ | $Y$ | $Z$ | $W$ |
| $X$ | | 1 | $Z$ | $Y$ | $W$ |
| $Y$ | | | 1 | $X$ | $W$ |
| $Z$ | | | | 1 | $W$ |
| $W$ | | | | | $1 \oplus X \oplus Y \oplus Z$ |

Thus, the simple objects have

$$\mathrm{FPdim}_{\mathcal{B}}(1) = \mathrm{FPdim}_{\mathcal{B}}(X) = \mathrm{FPdim}_{\mathcal{B}}(Y) = \mathrm{FPdim}_{\mathcal{B}}(Z) = 1, \quad \mathrm{FPdim}_{\mathcal{B}}(W) = 2,$$

and

$$\mathrm{FPdim}(\mathcal{B}) = 8.$$

We study unitary pre-MFCs. They have quantum

$$d_1 = d_X = d_Y = d_Z = 1, \quad d_W = 2,$$

and conformal dimensions[16]

$$(h_X, h_Y, h_Z, h_W) = (0, \frac{1}{2}, \frac{1}{2}, 0), (0, \frac{1}{2}, \frac{1}{2}, \frac{1}{8}), (0, \frac{1}{2}, \frac{1}{2}, \frac{1}{4}), (0, \frac{1}{2}, \frac{1}{2}, \frac{3}{8}),$$
$$(0, \frac{1}{2}, \frac{1}{2}, \frac{1}{2}), (0, \frac{1}{2}, \frac{1}{2}, \frac{5}{8}), (0, \frac{1}{2}, \frac{1}{2}, \frac{3}{4}), (0, \frac{1}{2}, \frac{1}{2}, \frac{7}{8}). \quad (\text{mod } 1)$$

The pre-MFCs have the $S$-matrix

$$\tilde{S} = \begin{pmatrix} 1 & 1 & 1 & 1 & 2 \\ 1 & 1 & 1 & 1 & 2 \\ 1 & 1 & 1 & 1 & -2 \\ 1 & 1 & 1 & 1 & -2 \\ 2 & 2 & -2 & -2 & 0 \end{pmatrix}.$$

We find $\{1, X\}$ are transparent. Note that, since the rank five is not a multiple of two, the pre-MFCs cannot be written as a Deligne tensor product of an MFC and $\{1, X\}$. Mixed-state TOs described by such pre-MFCs are called intrinsic in [9].

---

[16]One can find these conformal dimensions noticing that the pre-MFC is a rank five braided fusion subcategory of Ising $\boxtimes$ Ising [20].

The pre-MFCs have a nontrivial connected étale algebra

$$A \cong 1 \oplus X.$$

We condense this anyon. First, we assign

$$1 \to \underline{0},$$
$$X \to \underline{0}.$$

The fusion $X \otimes Y \cong Z$ reduces to

$$Y \cong Z.$$

The only nontrivial consistency condition imposed by the fusion rules are $W \otimes W$. This reduces to

$$W \otimes_A W \cong 2\underline{0} \oplus 2Y.$$

Since the RHS contains the vacuum, $W$ contains its dual. Furthermore, since the RHS has two vacua, $W$ should split into two invertible simple objects:

$$W \to W_1 \oplus W_2.$$

The fusion now reduced to

$$(W_1 \oplus W_2) \otimes_A (W_1 \oplus W_2) \cong 2\underline{0} \oplus 2Y.$$

Since $W^* \cong W$, there are two possibilities: 1) $W_{1,2}$ are self-dual, or 2) they are dual of each other. In the first case, we have

$$W_1 \otimes_A W_1 \cong \underline{0} \cong W_2 \otimes_A W_2.$$

This leads to

$$W_1 \otimes_A W_2 \cong Y \cong W_2 \otimes_A W_1.$$

In the second case, we have

$$W_1 \otimes_A W_2 \cong \underline{0} \cong W_2 \otimes_A W_1,$$

and

$$W_1 \otimes_A W_1 \cong Y \cong W_2 \otimes_A W_2.$$

We still need to fix $Y \otimes_A W_{1,2}$. The $Y \otimes W \cong W$ reduces to

$$Y \otimes_A (W_1 \oplus W_2) \cong W_1 \oplus W_2.$$

Logically, there are two possibilities, either $Y$ does not change $W_{1,2}$, or it swaps the two. By consistency, the first possibility is ruled out.[17] Therefore, in both cases, we have

$$Y \otimes_A W_1 \cong W_2, \quad Y \otimes_A W_2 \cong W_1.$$

In conclusion, we get four simple objects $\{\underline{0}, Y, W_1, W_2\}$ obeying $\mathbb{Z}_2 \times \mathbb{Z}_2$ fusion rules

| $\otimes_A$ | $\underline{0}$ | $Y$ | $W_1$ | $W_2$ |
|---|---|---|---|---|
| $\underline{0}$ | $\underline{0}$ | $Y$ | $W_1$ | $W_2$ |
| $Y$ | | $\underline{0}$ | $W_2$ | $W_1$ |
| $W_1$ | | | $\underline{0}$ | $Y$ |
| $W_2$ | | | | $\underline{0}$ |

when $W_{1,2}$ are self-dual, or $\mathbb{Z}_4$ fusion rules

| $\otimes_A$ | $\underline{0}$ | $Y$ | $W_1$ | $W_2$ |
|---|---|---|---|---|
| $\underline{0}$ | $\underline{0}$ | $Y$ | $W_1$ | $W_2$ |
| $Y$ | | $\underline{0}$ | $W_2$ | $W_1$ |
| $W_1$ | | | $Y$ | $\underline{0}$ |
| $W_2$ | | | | $Y$ |

when $W_1^* \cong W_2$. Namely,

$$\mathcal{B}_A \simeq \begin{cases} \mathrm{Vec}_{\mathbb{Z}_2 \times \mathbb{Z}_2} & (W_1^* \cong W_1, \ W_2^* \cong W_2), \\ \mathrm{Vec}_{\mathbb{Z}_4} & (W_1^* \cong W_2). \end{cases}$$

We can make the identifications more precise. As reviewed in the appendix, the symmetric center can be read off from the $S$-matrix. We find

$$\mathcal{B}' = \{1, X\}.$$

Since we condensed $A \in \mathcal{B}'$, we know $\mathcal{B}_A \simeq \mathcal{B}_A^0$. The resulting category of right $A$-modules is modular. With the help of the invariance of topological twist (A.15), we identify

$$\mathcal{B}_A \simeq \begin{cases} \mathrm{Vec}_{\mathbb{Z}_2}^{-1} \boxtimes \mathrm{Vec}_{\mathbb{Z}_2}^{-1} & (h_W = \frac{1}{4}, \frac{3}{4}), \\ \mathrm{ToricCode} & (h_W = 0, \frac{1}{2}), \quad (\mathrm{mod}\ 1) \\ \mathrm{Vec}_{\mathbb{Z}_4}^{\alpha} & (h_W = \frac{1}{8}, \frac{3}{8}, \frac{5}{8}, \frac{7}{8}). \end{cases} \tag{3.8}$$

---

[17]Here is a proof.
  Assume the opposite

$$Y \otimes_A W_1 \cong W_1, \quad Y \otimes_A W_2 \cong W_2.$$

When $W_{1,2}$ is self-dual, multiply $W_1$ from the right to the first fusion rule to get a contradiction $Y \cong \underline{0}$. When $W_1^* \cong W_2$, multiply $W_2$ from the right to the first fusion to obtain a contradiction $Y \cong \underline{0}$.

# A  Mathematical backgrounds

In this appendix, we collect relevant definitions and facts for our study. We assume physics-oriented readers. For more details, see the standard textbook [22].

## A.1  Definitions

Throughout the paper, $\mathcal{C}$ denotes a fusion category over the field $\mathbb{C}$ of complex numbers. Roughly speaking, it is a generalization of representations of groups. We know they are equipped with tensor products and direct sums. For example, the two-dimensional representation of $SU(2)$ obeys

$$2 \otimes 2 = 1 \oplus 3.$$

Similarly, a fusion category is equipped with fusion $\otimes$ and direct sum $\oplus$. (If necessary, we add the ambient category to subscripts of the binary operations, e.g., $\otimes_{\mathcal{C}}, \oplus_{\mathcal{C}}$.) In some sense, fusion category is simpler than the representations of $SU(2)$ because the number of simple objects – corresponding to irreducible representations of $SU(2)$ – is finite. The number is called the rank and denoted $\mathrm{rank}(\mathcal{C})$. The simple objects of $\mathcal{C}$ are denoted by $c_i$ with $i = 1, 2, \dots, \mathrm{rank}(\mathcal{C})$. Just like the trivial one-dimensional representation of $SU(2)$, by definition, fusion categories contain exactly one simple object called the unit $1 \in \mathcal{C}$ of the fusion obeying

$$\forall c \in \mathcal{C}, \quad 1 \otimes c \cong c \cong c \otimes 1. \tag{A.1}$$

The two isomorphisms

$$l_c : 1 \otimes c \xrightarrow{\sim} c, \quad r_c : c \otimes 1 \xrightarrow{\sim} c$$

are called left and right unit constraints (or unit isomorphisms). In a fusion category, all objects are direct sums of finitely many simple objects. (Compare with reducible representations which can be decomposed to direct sums of irreducible representations.) This property is called semisimple. All simple objects $c_i$'s admit their duals $c_i^*$. (Mathematically, such categories are called rigid.)

The fusion is characterized by fusion rules

$$c_i \otimes c_j \cong \bigoplus_{k=1}^{\mathrm{rank}(\mathcal{C})} N_{ij}{}^k c_k, \tag{A.2}$$

where $N_{ij}{}^k \in \mathbb{N}$ counts the number of $c_k$ in the fusion $c_i \otimes c_j$. Fusion of dual pairs $c_i, c_i^*$ contain exactly one unit:

$$c_i \otimes c_i^* \cong 1 \oplus \bigoplus_{c_k \not\cong 1} N_{c_i, c_i^*}{}^k c_k.$$

Given three objects $c, c', c'' \in \mathcal{C}$, there are two ways to compute their fusions. The result should be isomorphic

$$\alpha_{c,c',c''} : (c \otimes c') \otimes c'' \xrightarrow{\sim} c \otimes (c' \otimes c'').$$

The family of natural isomorphisms $\alpha$ is called the associator.

The $\mathbb{N}$-coefficients characterizing fusion assemble to $\text{rank}(\mathcal{C}) \times \text{rank}(\mathcal{C})$ matrices

$$(N_i)_{jk} := N_{ij}{}^k.$$

They are called fusion matrices. Since their elements are non-negative, we can apply the Perron-Frobenius theorem to get the largest positive eigenvalue $\text{FPdim}_{\mathcal{C}}(c_i)$ called Frobenius-Perron dimension of $c_i$ (or $N_i$). This quantifies the dimension of objects, while dimension of the ambient category is quantified by its squared sum

$$\text{FPdim}(\mathcal{C}) := \sum_{i=1}^{\text{rank}(\mathcal{C})} \left( \text{FPdim}_{\mathcal{C}}(c_i) \right)^2,$$

called the Frobenius-Perron dimension of $\mathcal{C}$.

In this paper, we are interested in spherical fusion category. In the category, another notion of dimension can be defined. They are called quantum dimensions and defined as the quantum (or categorical) trace

$$d_i := \text{tr}(a_{c_i}),$$

where $a : id_{\mathcal{C}} \cong (-)^{**}$ is called a pivotal structure. (Occasionally, we denote a quantum dimension of $c \in \mathcal{C}$ as $d_{\mathcal{C}}(c)$ to avoid confusion.) Similarly to the Frobenius-Perron dimension, its squared sum defines the categorical (or global) dimension of $\mathcal{C}$:

$$D^2(\mathcal{C}) := \sum_{i=1}^{\text{rank}(\mathcal{C})} d_i^2.$$

Note that there are two $D(\mathcal{C})$'s for a given categorical dimension, one positive and one negative. If $D^2(\mathcal{C}) = \text{FPdim}(\mathcal{C})$, $\mathcal{C}$ is called pseudo-unitary. If $\forall c_i \in \mathcal{C}$, $d_i = \text{FPdim}_{\mathcal{C}}(c_i)$, $\mathcal{C}$ is called unitary.

A braided fusion category (BFC) is a fusion category with a braiding $c$. (In order to avoid confusion, we denote a fusion category by $\mathcal{B}$ and simple objects by $b_i$ with $i = 1, 2, \ldots, \text{rank}(\mathcal{B})$.) A braiding is a family of natural isomorphisms

$$b, b' \in \mathcal{B}, \quad c_{b,b'} : b \otimes b' \xrightarrow{\sim} b' \otimes b.$$

A BFC is a pair $(\mathcal{B}, c)$ of fusion category $\mathcal{B}$ and a braiding $c$, but we usually just write $\mathcal{B}$. In our paper, we always assume BFCs are spherical. Spherical BFCs are called pre-modular fusion categories (pre-MFCs). Braidings are characterized by conformal dimensions $h_i$ of $b_i$. For example, double braiding is given by

$$c_{b_j, b_i} \cdot c_{b_i, b_j} \cong \sum_{k=1}^{\text{rank}(\mathcal{B})} N_{ij}{}^k \frac{e^{2\pi i h_k}}{e^{2\pi i (h_i + h_j)}} id_k, \tag{A.3}$$

where $id_k$ is the identity morphism at $b_k \in \mathcal{B}$. A simple object $b_i$ is called transparent if $\forall b_j \in \mathcal{B}$, $c_{b_j,b_i} \cdot c_{b_i,b_j} \cong id_{b_i \otimes b_j}$. By the unit property (A.1), the unit is always transparent. A collection of transparent simple objects is denoted $\mathcal{B}'$ or $Z_2(\mathcal{B})$ and called symmetric (or Müger) center. A BFC is called symmetric if all simple objects are transparent. A symmetric BFC $\mathcal{B}$ is called positive if $\forall b \in \mathcal{B}$, $d_b \in \mathbb{N}^\times$. The quantum trace of the double braiding defines the (unnormalized) $S$-matrix

$$\widetilde{S}_{i,j} := \mathrm{tr}(c_{b_j,b_i} \cdot c_{b_i,b_j}) = \sum_{k=1}^{\mathrm{rank}(\mathcal{B})} N_{ij}{}^k \frac{e^{2\pi i h_k}}{e^{2\pi i (h_i + h_j)}} d_k. \tag{A.4}$$

(We write $b_1 \cong 1$. Namely, the first row or column of $S$-matrices correspond to the unit object.) A normalized $S$-matrix is given by

$$S_{i,j} := \frac{\widetilde{S}_{i,j}}{D(\mathcal{B})}.$$

If the $S$-matrix is non-singular, i.e., there exists the inverse matrix, a BFC is called non-degenerate. A non-degenerate pre-MFC $\mathcal{B}$ is called modular or modular fusion category (MFC). Pure-state TOs are described by MFCs, and mixed-state TOs are described by (generically degenerate) pre-MFCs.

The conformal dimensions also define topological twists[18]

$$\theta_i := e^{2\pi i h_i}.$$

They form another modular matrix

$$T_{i,j} = \theta_i \delta_{i,j}.$$

Let $\mathcal{C}$ be a fusion category. An algebra in $\mathcal{C}$ is a triple $(A, \mu, u)$ of object $A \in \mathcal{C}$, and two morphisms $\mu : A \otimes A \to A$ and $u : 1 \to A$ called multiplication and unit morphisms, respectively. They are subject to two axioms: 1) associativity

$$\mu \cdot (id_A \otimes \mu) \cdot \alpha_{A,A,A} = \mu \cdot (\mu \otimes id_A), \tag{A.5}$$

and 2) unit axiom

$$id_A \cdot l_A = \mu \cdot (u \otimes id_A), \quad id_A \cdot r_A = \mu \cdot (id_A \otimes u). \tag{A.6}$$

---

[18]Mathematically, a twist is a natural isomorphism

$$\theta : id_\mathcal{B} \overset{\sim}{\Rightarrow} id_\mathcal{B}.$$

In the main text, we slightly abused the terminology, and called the coefficient as twist (as common in physics). In other words, we hided the identity morphism.

We usually denote an algebra by $A$. An algebra $A$ is called connected if $\dim_{\mathbb{C}} \mathcal{C}(1, A) = 1$. An algebra $A$ is called separable if $\exists \widetilde{\mu} : A \to A \otimes A$ such that $\widetilde{\mu} \cdot \mu = id_A$. An algebra $A$ in a BFC $(\mathcal{B}, c)$ is called commutative if

$$\mu \cdot c_{A,A} = \mu. \tag{A.7}$$

A commutative separable algebra is called an étale algebra.

Let $\mathcal{C}$ be a fusion category and $A \in \mathcal{C}$ an algebra. A right $A$-module is a pair $(m, p)$ of object $m \in \mathcal{C}$ and a morphism $p : m \otimes A \to m$. The morphism can be viewed as a right action of $A$ on $m$ (so the name). They are subject to the associativity

$$p \cdot (p \otimes_{\mathcal{C}} id_A) = p \cdot (id_m \otimes_{\mathcal{C}} \mu) \cdot \alpha_{m,A,A}, \tag{A.8}$$

and unit axiom

$$p \cdot (id_m \otimes_{\mathcal{C}} u) = id_m \cdot r_m. \tag{A.9}$$

We usually denote a right $A$-module by just $m$. They form the category $\mathcal{C}_A$ of right $A$-modules. If $\mathcal{C}$ admits a braiding, it contains an important subcategory. Let $\mathcal{B}$ be a BFC and $A \in \mathcal{B}$ an algebra. A right $A$-module $(m, p)$ is called dyslectic (or local) [23] if

$$p \cdot c_{A,m} \cdot c_{m,A} = p. \tag{A.10}$$

They form the category $\mathcal{B}_A^0$ of dyslectic right $A$-modules, which is by construction a subcategory of $\mathcal{B}_A$. In the anyon condensation [11, 12, 13] in pure-state TOs, $\mathcal{B}_A, \mathcal{B}_A^0$ are called confined and deconfined phase, respectively.

## A.2 Facts

First, since we assume our pre-MFCs are degenerate, we need to check an existence of nontrivial transparent simple object. A useful tool to judge (non-)degeneracy of BFCs is the monodromy charge matrix [24, 25, 26]

$$M_{i,j} := \frac{\widetilde{S}_{i,j} \widetilde{S}_{1,1}}{\widetilde{S}_{1,i} \widetilde{S}_{1,j}} = \frac{S_{i,j} S_{1,1}}{S_{1,i} S_{1,j}}. \tag{A.11}$$

We have [8]

$$b_i \in \mathcal{B}' \iff \forall b_j \in \mathcal{B}, \quad M_{i,j} = 1.$$

If there exists a nontrivial simple object $b_i$ giving $M_{i,j} = 1$ for all $b_j \in \mathcal{B}$, then the BFC is degenerate.[19]

---

[19]This tool was also used in [27, 28, 29, 30] to give a mathematical explanation when and why which symmetries emerge.

Let $\mathcal{B}$ be a pre-MFC and $A \in \mathcal{B}$ a connected étale algebra.[20] The separability of $A \in \mathcal{B}$ is equivalent to the semisimplicity of $\mathcal{B}_A$ [31, 22]. Hence, in our setup, $\mathcal{B}_A$ is also a fusion category [31, 22]. Its fusion is denoted $\otimes_A$. The fusion category $\mathcal{B}_A$ has [32, 33]

$$\mathrm{FPdim}(\mathcal{B}_A) = \frac{\mathrm{FPdim}(\mathcal{B})}{\mathrm{FPdim}_\mathcal{B}(A)}. \tag{A.12}$$

The Frobenius-Perron dimension of an algebra $A \in \mathcal{B}$

$$A \cong \bigoplus_{i=1}^{\mathrm{rank}(\mathcal{B})} n_i b_i$$

with $n_i \in \mathbb{N}$ is given by

$$\mathrm{FPdim}_\mathcal{B}(A) = \sum_{i=1}^{\mathrm{rank}(\mathcal{B})} n_i \mathrm{FPdim}_\mathcal{B}(b_i).$$

A useful tool in identifying $\mathcal{B}_A$ is the free module functor defined by

$$F_A := - \otimes A : \mathcal{B} \to \mathcal{B}_A, \tag{A.13}$$

sending $b \in \mathcal{B}$ to $b \otimes A$. The functor is a surjective (or dominant) tensor functor [34, 33]. Since it is surjective, $\forall m \in \mathcal{B}_A$, $\exists b_j \in \mathcal{B}$ such that $m$ is a subobject of $F_A(b_j)$. Furthermore, since it is tensor, we have preservation of Frobenius-Perron dimensions

$$\forall b \in \mathcal{B}, \quad \mathrm{FPdim}_\mathcal{B}(b) = \mathrm{FPdim}_{\mathcal{B}_A}(F_A(b)),$$

and

$$\forall b, b' \in \mathcal{B}, \quad F_A(b) \otimes_A F_A(b') \cong F_A(b \otimes b').$$

When $b \otimes b'$ is a (finite) direct sum of simple objects, we can distribute employing the exactness of the fusion:

$$(b_1 \oplus b_2 \oplus \cdots \oplus b_n) \otimes A \cong (b_1 \otimes A) \oplus (b_2 \otimes A) \oplus \cdots \oplus (b_n \otimes A).$$

Using these properties, one can find all simple objects of $\mathcal{B}_A$, and compute their fusions $\otimes_A$. A right $A$-module $m \in \mathcal{B}_A$ has

$$d_{\mathcal{B}_A}(m) = \frac{d_\mathcal{B}(m)}{d_\mathcal{B}(A)}. \tag{A.14}$$

The subcategory $\mathcal{B}_A^0 \subset \mathcal{B}_A$ admits braiding [23]. In our setup, $\mathcal{B}_A$ and hence $\mathcal{B}_A^0$ are spherical [34]. Thus, $\mathcal{B}_A^0$ is another pre-MFC describing another mixed-state TO. Note that

---

[20]A systematic method to classify connected étale algebras in (pre-)MFCs has recently been developed in [15, 16, 18, 19, 20]. We use the method to find connected étale algebras.

$\mathcal{B}_A \simeq \mathcal{B}_A^0$ if $A \in \mathcal{B}$ is transparent because $c_{A,m} \cdot c_{m,A} \cong id_{m \otimes A}$. Thus, in this case, $\mathcal{B}_A$ also admits braiding. We summarize these facts in the following

| Notation | Mathematical meaning | Physical meaning |
|:---:|:---:|:---:|
| $\mathcal{B}$ | Pre-MFC | Mixed-state TO |
| $A \in \mathcal{B}$ | Connected étale algebra | Condensable anyon . |
| $\mathcal{B}_A$ | Spherical FC | Confined phase |
| $\mathcal{B}_A^0$ | Pre-MFC | Deconfined phase |

**Table 4:** Dictionary

A useful fact to fix $\mathcal{B}_A^0$ is the invariance of topological twists [34, 35]

$$e^{2\pi i h_b^{\mathcal{B}}} = e^{2\pi i h_b^{\mathcal{B}_A^0}}, \tag{A.15}$$

where $h_b^{\mathcal{B}}$ is a conformal dimension of $b \in \mathcal{B}$ and $h_b^{\mathcal{B}_A^0}$ is that of $b \in \mathcal{B}_A^0$. This guarantees deconfined particles have the same conformal dimensions (mod 1) as in the ambient category.

Finally, let us recall an interesting fact. In our examples, we find some anyon condensations in mixed-state TOs give pure-state TOs. Mathematically, it is known when this happens. Let $\mathcal{B}$ be a pre-MFC and $A \in \mathcal{B}$ a connected étale algebra. Condensation of such an anyon gives a surjective functor $F : \mathcal{B} \to \mathcal{M}$ to an MFC $\mathcal{M}$. The functor is called a modularization of $\mathcal{B}$. We have [36, 37]

$$\exists \text{modularization of } \mathcal{B} \iff \forall b_j \in \mathcal{B}', \quad \theta_{b_j} = 1. \tag{A.16}$$

A regular algebra giving $(\mathcal{B}')_A \simeq \text{Vect}_{\mathbb{C}}$ provides the modularization $F_A : \mathcal{B} \to \mathcal{B}_A$ [38]. Physically, such a (generically composite) anyon consists of all transparent simple anyons. If all transparent anyons are bosons, condensation of them makes $\mathcal{B}'$ 'trivial,' resulting in an MFC describing pure-state TO. The resulting (super-)MFC is nothing but the topological invariant $\mathcal{A}^{\min}$ of equivalence classes of mixed-state TOs. In particular, it is known [21, 22] that positive symmetric BFCs $\mathcal{B}$'s admit modularization $F_A : \mathcal{B} \to \text{Vect}_{\mathbb{C}}$. (Therefore, such mixed-states TOs belong to the same equivalence class.) The simplest pre-MFC which fails to admit modularization is the $\mathbb{Z}_2$ pre-MFC with a transparent fermion $f \otimes f \cong 1, \theta_f = -1$. (Mathematically, the pre-MFC is denoted sVec.) The pre-MFC is the $\mathbb{Z}_2$ subcategory of the Ising MFC.

## A.3  Mathematical anyon condensation

In this subsection, we redo anyon condensations presented in the introduction from a mathematical viewpoint. For more details, see [20].

As in the introduction, we take $\mathcal{B} \simeq \mathrm{Rep}(S_3)$ as our old pre-MFC. It has three simple objects $\{1, X, Y\}$ obeying fusion rules

| $\otimes$ | $1$ | $X$ | $Y$ |
|---|---|---|---|
| $1$ | $1$ | $X$ | $Y$ |
| $X$ | | $1$ | $Y$ |
| $Y$ | | | $1 \oplus X \oplus Y$ |

.

We consider condensation of three connected étale algebras

$$1 \oplus X, 1 \oplus Y, 1 \oplus X \oplus 2Y \in \mathrm{Rep}(S_3).$$

$1 \oplus X \in \mathbf{Rep}(S_3)$. The free module functor $F_A$ gives

$$F_A(1) \cong 1 \oplus X \cong F_A(X), \quad F_A(Y) \cong 2Y.$$

Since the functor is surjective, simple objects of $\mathcal{B}_A$ are contained in these two. As the functor preserves Frobenius-Perron dimensions, they have

$$\mathrm{FPdim}_{\mathcal{B}_A}(F_A(1)) = 1 = \mathrm{FPdim}_{\mathcal{B}_A}(F_A(X)), \quad \mathrm{FPdim}_{\mathcal{B}_A}(F_A(Y)) = 2.$$

Note that the latter has Frobenius-Perron dimension two. Thus, it can be a direct sum of two simple objects with Frobenius-Perron dimensions one. Is this the case? In order to answer this point, we compute the Frobenius-Perron dimension of $\mathcal{B}_A$. It is given by (A.12):

$$\mathrm{FPdim}(\mathcal{B}_A) = \frac{6}{2} = 3.$$

If $F_A(Y)$ were simple, it would contribute to the Frobenius-Perron dimension by $2^2$, which exceeds three, a contradiction. Thus, we learn $F_A(Y)$ should be a direct sum of two simple objects with Frobenius-Perron dimensions one. In other words, $\mathcal{B}_A$ has

$$m_1 \cong 1 \oplus X, \quad m_2 \cong Y \cong m_3.$$

This is the mathematical meaning of "$Y$ splits into two." They have

$$\mathrm{FPdim}_{\mathcal{B}_A}(m_1) = \mathrm{FPdim}_{\mathcal{B}_A}(m_2) = \mathrm{FPdim}_{\mathcal{B}_A}(m_3) = 1.$$

This matches the Frobenius-Perron dimension

$$1^2 + 1^2 + 1^2 = 3 = \mathrm{FPdim}(\mathcal{B}_A).$$

To fix $\mathcal{B}_A$, we compute new fusion rules. Here, we again use the free module functor. Since it is tensor, we can compute new fusion rules from the old ones. For example,

$$\begin{aligned}
m_1 \otimes_A m_1 =& F_A(1) \otimes_A F_A(1) \\
\cong& F_A(1 \otimes 1) \\
\cong& F_A(1) \cong m_1.
\end{aligned}$$

Similar computation shows $m_1$ is the unit of $\mathcal{B}_A$. What is left is fusion rules among $m_{2,3}$. We have

$$\begin{aligned}
(m_2 \oplus m_3) \otimes_A (m_2 \oplus m_3) =& F_A(Y) \otimes_A F_A(Y) \\
\cong& F_A(Y \otimes Y) \\
\cong& F_A(1 \oplus X \oplus Y) \cong 2m_1 \oplus m_2 \oplus m_3.
\end{aligned}$$

Here, since $(2Y)^* \cong 2Y$, there are two logical possibilities: i) $m_{2,3}$ are self-dual, or ii) they are dual to each other, $m_2^* \cong m_3$. In the first case, we have $m_2 \otimes_A m_2 \cong m_1 \cong m_3 \otimes_A m_3$. In the second case, we have $m_2 \otimes_A m_3 \cong m_1 \cong m_3 \otimes_A m_2$. We find the second case is correct. To show this, assume the first case. Then, we must have

$$m_2 \otimes_A m_3 \cong m_2, \quad m_3 \otimes_A m_2 \cong m_3,$$

or

$$m_2 \otimes_A m_3 \cong m_3, \quad m_3 \otimes_A m_2 \cong m_2.$$

Both of them are inconsistent: for the first case, multiply $m_2$ from the right to get $m_3 \cong m_1$, a contradiction; for the second case, multiply $m_3$ from the right to get $m_2 \cong m_1$, a contradiction. Thus, $m_{2,3}$ are dual to each other:

$$m_2 \otimes_A m_3 \cong m_1 \cong m_3 \otimes_A m_2.$$

We still have two logical possibilities:

$$m_2 \otimes_A m_2 \cong m_2, \quad m_3 \otimes_A m_3 \cong m_3,$$

or

$$m_2 \otimes_A m_2 \cong m_3, \quad m_3 \otimes_A m_3 \cong m_2.$$

We find the second is correct. To prove this, assume the first. Then, multiply $m_3$ from the right to get $m_2 \cong m_1$, a contradiction. This fixes all fusion rules. To summarize, after condensing $1 \oplus X \in \mathrm{Rep}(S_3)$, we have rank three pre-MFC $\mathcal{B}_A = \{m_1, m_2, m_3\}$ with fusion rules

| $\otimes_A$ | $m_1$ | $m_2$ | $m_3$ |
|---|---|---|---|
| $m_1$ | $m_1$ | $m_2$ | $m_3$ |
| $m_2$ | | $m_3$ | $m_1$ |
| $m_3$ | | | $m_2$ |

One identifies

$$\mathcal{B}_A \simeq \mathrm{Vec}_{\mathbb{Z}_3}.$$

$1 \oplus Y \in \mathbf{Rep}(S_3)$. Since we have explained the previous condensation in detail, we will be brief below. The free module functor gives

$$F_A(1) \cong 1 \oplus Y, \quad F_A(X) \cong X \oplus Y, \quad F_A(Y) \cong (1 \oplus Y) \oplus (X \oplus Y).$$

This suggests simple objects

$$m_1 \cong 1 \oplus Y, \quad m_2 \cong X \oplus Y.$$

Indeed, this matches Frobenius-Perron dimension

$$\mathrm{FPdim}(\mathcal{B}_A) = \frac{6}{3} = 2.$$

Since $m_1$ is the unit, what is left is to compute $m_2 \otimes_A m_2$. This can be obtained as

$$\begin{aligned} m_2 \otimes_A m_2 =& F_A(X) \otimes_A F_A(X) \\ \cong& F_A(X \otimes X) \\ \cong& F_A(1) \cong m_1. \end{aligned}$$

Therefore, the condensed theory has rank two $\mathcal{B}_A = \{m_1, m_2\}$ with fusion rules

| $\otimes_A$ | $m_1$ | $m_2$ |
|---|---|---|
| $m_1$ | $m_1$ | $m_2$ |
| $m_2$ | | $m_1$ |

One identifies

$$\mathcal{B}_A \simeq \mathrm{Vec}_{\mathbb{Z}_2}.$$

$1 \oplus X \oplus 2Y \in \mathbf{Rep}(S_3)$. The free module functor gives

$$F_A(1) \cong 1 \oplus X \oplus 2Y \cong F_A(X), \quad F_A(Y) \cong 2(1 \oplus X \oplus 2Y).$$

The result suggests the only simple object is

$$m_1 \cong 1 \oplus X \oplus 2Y,$$

and this matches the Frobenius-Perron dimension

$$\mathrm{FPdim}(\mathcal{B}_A) = \frac{6}{6} = 1.$$

The condensed theory is identified as

$$\mathcal{B}_A \simeq \mathrm{Vect}_{\mathbb{C}}.$$

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
