# Peer review of "Anyon condensation in mixed-state topological order"

_SciPost Physics Core_

## Round 3 · Referee Report · Anonymous (Referee 1) · 2025-7-29

Strengths

  1. The paper addresses an interesting question (what happens when anyons condense in mixed states rather than pure states?)

  2. The paper illustrates the discussion with plenty of examples

Weaknesses

Even after the revision, the presentation is still misleading: 1- The authors now call their first result' aphysical theorem'. There is no such thing. Theorem are statements whose validity is established by a proof . Physical truth' can be established by mathematical proof or by empirical evidence. Neither is provided for thephysical theorem' in this paper. It is a conjecture backed up by a general discussion and some examples, so it should be called a conjecture.

2- The status of the other theorems is also not clearly stated. The authors now cite other papers, suggesting that the theorems were fully formulated and proved in those papers. However, they introduce the theorems by saying `We clarified'. What exactly is the clarification here? Is it the application to physics, and the illustration with examples? That would be acceptable, but should be stated clearly.

Report

As explained in my first report, the paper addresses an interesting question (how the discussion of topological order as resulting from the breaking of higher order symmetry is changed when the symmetry breaking state is mixed rather than pure). It collects relevant mathematical results, adds a conjecture and illustrates all of those with examples. I find this interesting, and worth publishing. However, the presentations of the results is still misleading. I suggest changes below.

Requested changes

1- The `physical theorem' should be called Conjecture

2- In the introductory paragraphs of the other three theorems the authors should state clearly what they are contributing in this paper: `We discuss and illustrate the following theorem' or similar.

Recommendation

Ask for minor revision

---

## Round 3 · List of Changes

1. For the first comment, we put the superscript ph on the first theorem to emphasize this is not a mathematical theorem but a physical result and for the other three theorems we cited math papers.
  2. For the second comment, we shortened the paper by cutting one appendix.

---

## Editorial Decision

resubmitted